# The biomechanical role of extra-axonemal structures in shaping the flagellar beat of *Euglena gracilis*

Giancarlo Cicconofri[1], Giovanni Noselli[1], Antonio DeSimone[1,2]*

[1]SISSA - International School for Advanced Studies, Trieste, Italy; [2]The BioRobotics Institute, Scuola Superiore Sant'Anna, Trieste, Italy

**Abstract** We propose and discuss a model for flagellar mechanics in *Euglena gracilis*. We show that the peculiar non-planar shapes of its beating flagellum, dubbed 'spinning lasso', arise from the mechanical interactions between two of its inner components, namely, the axoneme and the paraflagellar rod. The spontaneous shape of the axoneme and the resting shape of the paraflagellar rod are incompatible. Thus, the complex non-planar configurations of the coupled system emerge as the energetically optimal compromise between the two antagonistic components. The model is able to reproduce the experimentally observed flagellar beats and the characteristic geometric signature of spinning lasso, namely, traveling waves of torsion with alternating sign along the length of the flagellum.

## Introduction

Flagella and cilia propel swimming eukaryotic cells and drive fluids on epithelial tissues of higher organisms (*Alberts et al., 2015*). The inner structure of the eukaryotic flagellum is an arrangement of microtubules (MTs) and accessory proteins called the axoneme (Ax). A highly conserved structure in evolution, the Ax typically consists of nine cylindrically arranged MT doublets cross-bridged by motor proteins of the dynein family. An internal central pair of MTs is connected by radial spokes to the nine peripheral doublets, determining the typical '9+2' axonemal structure. Motor proteins hydrolyze ATP to generate forces that induce doublet sliding. Due to mechanical constraints exerted by linking proteins (nexins) and the basal body, dynein-induced sliding of MTs translates into bending movements of the whole structure. Motor proteins are thought to self regulate their activity via mechanical feedback, generating the periodic beats of flagella, see, for example, *Brokaw, 2009* and *Lindemann and Lesich, 2010*.

Despite a general consensus on the existence of a self-regulatory mechanism, the inner working of the Ax is not fully understood and it is still the subject of active research (*Wan and Jékely, 2020*). While bending-through-sliding is the accepted fundamental mechanism of flagellar motility, how specific flagellar shapes are determined is not yet clear. Nodal cilia present in early embryonic development display markedly non-planar beats (*Buceta et al., 2005*). On the other hand, for the most studied swimming microorganisms, such as animal sperm cells and the biflagellate alga *Chlamydomonas reinhardtii*, the flagellar beat is, to a good approximation, planar. For these organisms, beat planarity is thought to be induced by the inter-doublet links between one pair of MTs, typically those numbered 5 and 6 (*Lin et al., 2012*). These links inhibit the relative sliding of the 5-6 MTs pair, thus selecting a beating plane that passes through the center of the Ax and the midpoint between the inhibited MTs.

A remarkable deviation from the flagellar structure of the aforementioned organisms is found in euglenids and kinetoplastids. These flagellated protists have an extra element attached alongside the Ax (*Cachon et al., 1988*), a slender structure made of a lattice-like arrangement of proteins

*For correspondence:
desimone@sissa.it

Competing interests: The authors declare that no competing interests exist.

called 'paraxial' or 'paraflagellar' rod (PFR), see *Figure 1*. The latter name is more common, but the former is possibly more accurate (*Rosati et al., 1991*). PFRs are attached via bonding links to up to four axonemal MTs, depending on the species (*Walne and Dawson, 1993*). PFRs are thought to be passive but, at least in the case of *Euglena gracilis*, some degree of activity is not completely ruled out (*Piccinni, 1975*).

*E. gracilis* has two flagella, designated as dorsal and ventral. The ventral flagellum remains within the reservoir, an invaginated region of the cell. The dorsal, PFR-bearing, flagellum emerges from the reservoir and serves as a propulsive apparatus by means of periodic beating. In this paper, we show that the beating style of *E. gracilis*, sometimes dubbed 'spinning lasso' (*Bovee, 1982*), is characterized by a distinctive geometric signature, namely, traveling torsional peaks with alternating sign along the length of the flagellum. Moreover, we put forward and test the hypothesis that this distinctive beating style arises from the PFR-Ax mechanical interaction.

In order to put our hypothesis into context, we observe that the flagellar beat of PFR-bearing kinetoplastid organisms, such as *Leishmania* and *Crithidia*, is planar (*Gadelha et al., 2007*). An apparent exception to beat planarity in kinetoplastids is found in the pathogenic parasite *Trypanosoma brucei*, which shows a characteristic non-planar 'drill-like' motion (*Langousis and Hill, 2014*). It has been claimed that the flagellar structure alone could account for the emergence this motion (*Koyfman et al., 2011*). However, the flagellum of *T. brucei* is not free, like that of *Leishmania* and *Crithidia*, but it is attached to the organism for most of its length, wrapped helically around the cell body. According to *Alizadehrad et al., 2015*, the flagellum-body mechanical interaction can alone explain *T. brucei*'s distinctive motion. Confirming this conclusion, *Wheeler, 2017* showed that *T. brucei* mutants with body-detached flagellum generate fairly planar beating. It is conjectured that the PFR-Ax bonds operate as the 5-6 interdoublet links in *Chlamydomonas* and sperm cells, inhibiting MTs sliding and selecting a plane of beat (*Woolley et al., 2006*).

The spinning lasso beat of *E. gracilis* does not conform to this scenario. Indeed, *E. gracilis* beating style is characterized by high asymmetry and non-planarity. The full 3d flagellar kinematics of freely swimming cells has recently been revealed by *Rossi et al., 2017* thanks to a mixed approach based on hydrodynamic theory and image analysis. As we report in the first part of this paper, the geometry of the spinning lasso is characterized by traveling waves of torsion with alternating sign along the flagellum length.

We argue that the key to the emergence of non-planarity lies in a prominent structural asymmetry of the PFR-Ax attachment in euglenid flagella. *Figure 1* shows a sketch of the cross-section of the euglenid flagellum redrawn from the electron microscopy studies by *Melkonian et al., 1982* and *Bouck et al., 1990*. Following the latter studies, we number MTs in increasing order in the clockwise direction, as seen from the distal end of the Ax. Notice that a different convention is commonly used in structural studies of cilia and flagella, see, for example, *Lin and Nicastro, 2018*. The PFR is attached to MTs 1, 2, and 3. We consider two lines. One line (dashed) passes through the center of the Ax and MT 2, in the middle of the bonding complex. The other (solid) line connects the center of the Ax and the center of the PFR. The two lines cross each other. This is the structural feature on which we build our model.

In modeling the flagellar complex, we assume that the bonding links to the PFR select the local spontaneous beating plane of the Ax, from the same principle of MTs' sliding inhibition discussed above. The local spontaneous beating plane so generated passes through the dashed line in *Figure 1*. We follow closely *Hilfinger and Jülicher, 2008* and *Sartori et al., 2016a* in our modeling of the Ax, while we use a simple elastic spring model for the PFR. We show that, under generic actuation, the two flagellar components cannot be simultaneously in their respective states of minimal energy, and this crucially depends on the offset between the spontaneous beating plane of the Ax (dashed line in *Figure 1*) and the line joining the PFR-Ax centers (solid line in *Figure 1*). Instead, the typical outcome is an elastically frustrated configuration of the system, in which the two competing components drive each other out of plane. Under dyneins activation patterns that, in the absence of extra-axonemal structures, would produce an asymmetric beat similar to those of *Chlamydomonas* (*Qin et al., 2015*), or *Volvox* (*Sareh et al., 2013*), the model specifically predicts the torsional signature of the spinning lasso.

Interestingly, the lack of symmetry of the spinning lasso beat produces swimming trajectories with rotations coupled with translations (*Rossi et al., 2017*). In turn, cell body rotations have a key role in the light-guided navigation behavior of phototactic unicellular organisms (*Goldstein, 2015*),

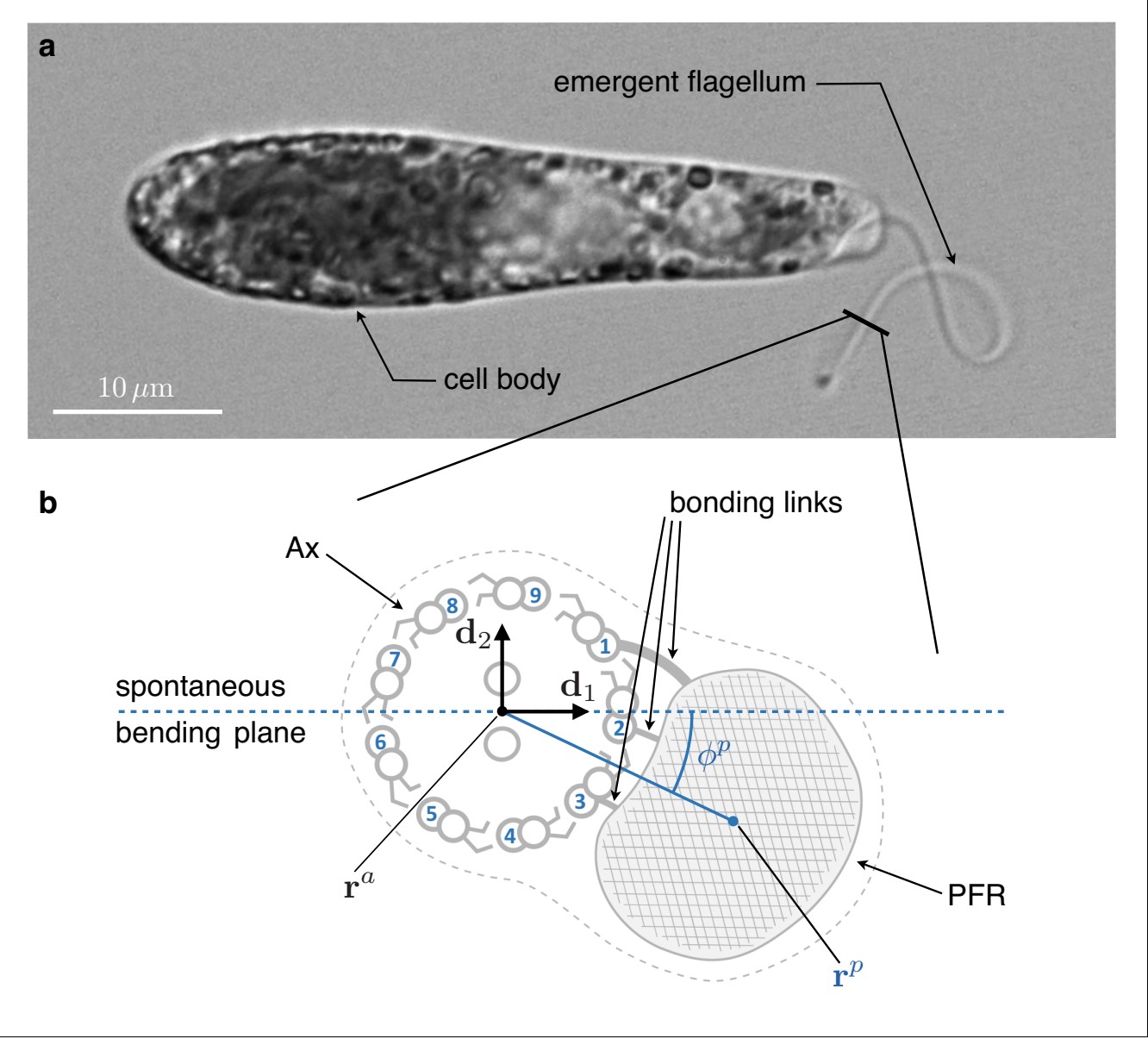

**Figure 1.** Inner structure of Euglena gracilis' flagellum. (**a**) A specimen of freely swimming *Euglena gracilis*, and (**b**) a sketch of the cross-section of its flagellum, as seen from the distal end. The flagellar inner structure is composed by the paraflagellar rod (PFR, textured), and the axoneme (Ax). The PFR is connected via bonding links to the axonemal doublets 1, 2, and 3. The inner structure of the flagellum is enclosed by the flagellar membrane (dotted contour). By inhibiting MTs' sliding, the PFR selects the spontaneous bending plane of the Ax (dashed line). As a key geometric feature, the solid line that joins the Ax center $\mathbf{r}^a$ and the PFR center $\mathbf{r}^p$ crosses at an angle $\phi^p$ the spontaneous bending plane. Doublets are numbered following the convention adopted in the electron microscopy studies *Melkonian et al., 1982* and *Bouck et al., 1990*, to facilitate comparison. The opposite convention, in which microtubules are numbered in increasing order in the anti-clockwise direction when seen from the distal end of the Ax, is far more common in structural studies of cilia.

and of *E. gracilis* in particular, which has recently attracted renewed attention (*Giometto et al., 2015*; *Ogawa et al., 2016*; *Tsang et al., 2018*). Rotations along the major axis of the cell body produced by the spinning lasso beat allow the light-sensing apparatus of the organism to constantly scan the environment, and align *E. gracilis* with light intensity gradients.

In light of these observations, our analysis shows that the beat of the euglenid flagellum can be seen as an example of a biological function arising from the competition between antagonistic structural components. It is not dissimilar from the body-flagellum interaction in *T. brucei*, which

generates 3d motility. But the principle is much more general in biology and many other examples can be found across kingdoms and species, and at widely different scales. For instance in plants, a mechanism of seed dispersal arises from the mechanical competition between the two valves of the seed pods, see, for example *Armon et al., 2011* for *Bauhinia variegata* and *Hofhuis et al., 2016* for *Cardamine hirsuta*. Contraction by antagonistic muscles is key for animal movement and, in particular, for the functioning of hydrostatic skeletons (used from wormlike invertebrates to arms and tentacles of cephalopods, to the trunk of elephants, see *Kier, 2012*). The mechanical coupling of the helical periplasmic flagella to the rod-shaped cell cylinder determines the flat-wave morphology of the Lyme disease spirochete *Borrelia burgdorferi* (*Dombrowski et al., 2009*). Antagonistic contraction along perpendicularly oriented families of fibres is at work at the subcellular level, for example in the antagonistic motor protein dynamics in contractile ring structures important in eukaryotic cell division and development, see, for example *Coffman et al., 2016*. At the same subcellular scale, competing elastic forces arising from lipid-protein interactions are often crucial in determining the stability of complex shapes of the cellular membrane (*Moser von Filseck et al., 2020*), and in the case of the overall structure of the coronavirus envelope (*Schoeman and Fielding, 2019*).

## Observations

We first analyze the experimental data from the 3d reconstruction of the beating euglenid flagellum obtained in *Rossi et al., 2017* for freely swimming organisms. Swimming *E. gracilis* cells follow generalized helical trajectories coupled with rotation around the major axis of the cell body. It is precisely this rotation that allows for a 3d reconstruction of flagellar shapes from 2d videomicroscopy images. *E. gracilis* takes many beats to close one complete turn around its major axis. So, while rotating, cells expose their flagellar beat to the observer from many different sides. Stereomatching techniques can then be employed to reconstruct the flagellar beat in full (assuming periodicity and regularity of the beat). *Figure 2* shows $N = 10$ different curves in space describing the euglenid flagellum in different instants within a beat taken from *Rossi et al., 2017*. The reconstruction fits well experimental data from multiple specimens. The figure also illustrates the computed torsion of the flagellar curve at each instant (not previously published). Torsion, the rate of change of the binormal vector, is the geometric quantity that measures the deviation of a curve from a planar path (see the 'Results' Section below for the formal definition). The 'spinning lasso' exhibits here a distinct torsional signature: torsion peaks of alternate sign that travel from the proximal to the distal end of the flagellum. We return to this point below.

To further investigate *E. gracilis*' flagellar beat, we observed stationary cells trapped at the tip of a capillary. In this setting, the flagellum is not perturbed by the hydrodynamic forces associated with *E. gracilis*' rototranslating swimming motion. The beat can then manifest itself in its most 'pristine' form. We recorded trapped cells during periodic beating. We rotated the capillary and recorded the same beating cell from different viewpoints. Videomicroscopy images from one specimen are shown in *Figure 3* and *Video 1*. While with fixed specimens we cannot reconstruct reliably the 3d flagellar shapes, *Figure 3* shows that there is a high stereographical consistency with the flagellar shapes obtained from swimming organisms. Flagellar non-planarity is thus not intrinsically associated with swimming, which reinforce the idea that the mechanism that generates non-planar flagellar shapes might be structural in origin. Moreover, these observations justify the choice we made in our study to focus on a model of flagellar mechanics for stationary organisms, allowing for substantial simplifications.

Getting back to the torsion measurement in *Figure 2*, we show here that the pattern of torsional peaks of alternate sign is consistent with *E. gracilis*' flagellar shapes as seen from common 2d microscopy, for either swimming or trapped organisms. Typically, the 2d outline (i.e. the projection on the focal plane) of a beating euglenid flagellum is that of a looping curve, see *Figure 3* and, for example *Tsang et al., 2018* for independent observations. Consider now an idealized 3d model of the spinning lasso geometry: a 'torsion dipole'. This simple geometric construction, shown in *Figure 3*, consists of a curve with two singular points of concentrated torsion with opposite sign. If we move along the curve, from proximal end to distal end, we first remain on a fixed plane (blue). Then the plane of the curve abruptly rotates by 90° (red plane) first, and then back by 90° in the opposite direction (yellow plane). These abrupt changes correspond to concentrated torsional peaks of opposite sign. When seen in a two-dimensional projection, the torsion dipole generates a looping curve, which closely matches euglenid flagella's outlines during a spinning lasso beat.

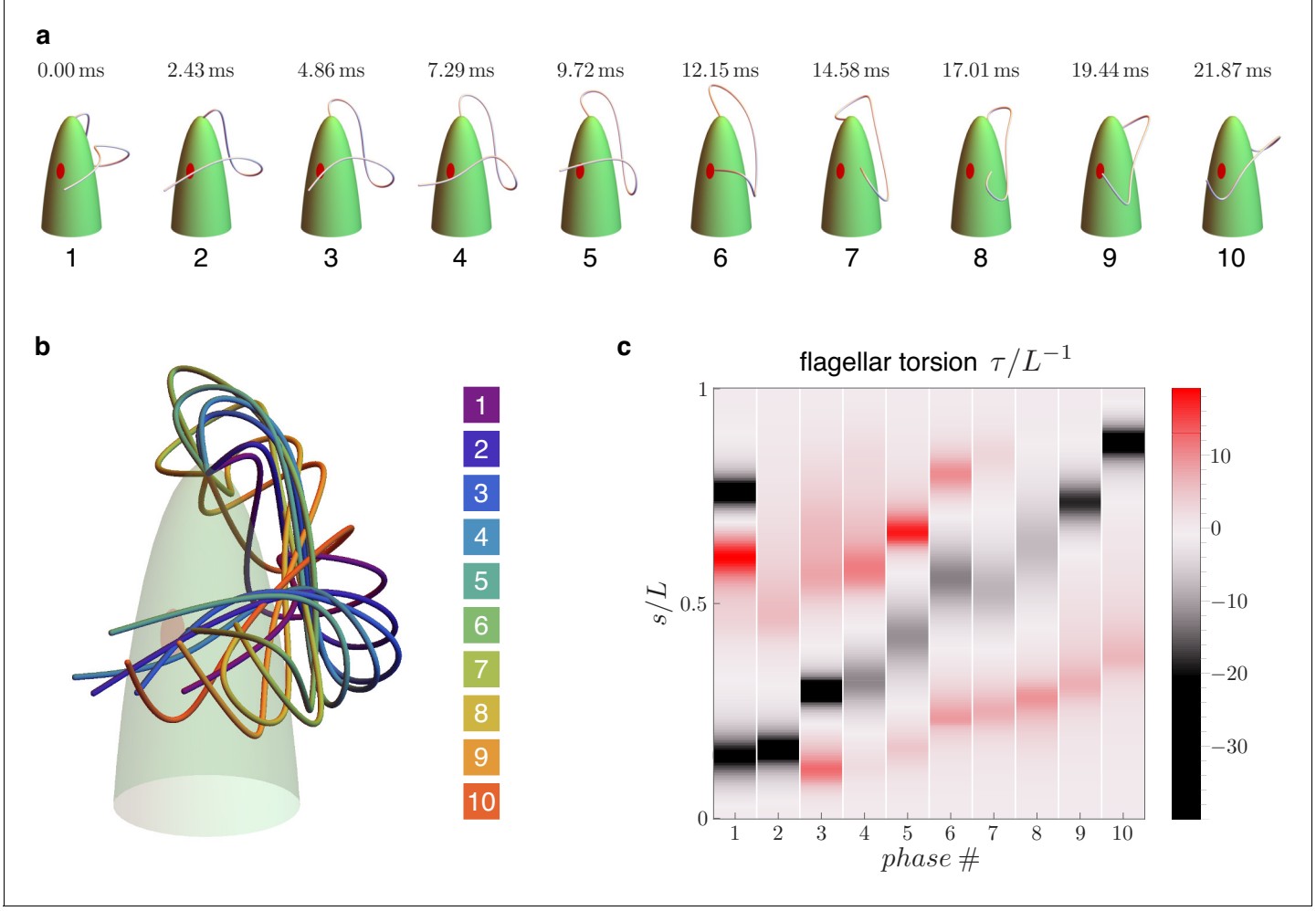

**Figure 2.** Flagellar beat kinematics of freely swimming *Euglena gracilis*. (**a**) $N = 10$ flagellar configurations in evenly spaced instants (phases) within a periodic beat. (**b**) The same configurations overlapped and color coded according to their phases. (**c**) Computed torsion $\tau = \tau(s)$ as a function of the flagellar arc length $s$. The plot is presented in terms of the normalized quantities $\tau/L^{-1}$ and $s/L$, where $L$ is the total length of the flagellum. Panels (**a-b**) are adapted from Figure 5.E of *Rossi et al., 2017*.

The online version of this article includes the following source code for figure 2:

**Source code 1.** Experimental flagellar waveforms and torsion calculator.

## Mechanical model

We model Ax and PFR as cylindrical structures with deformable centerlines, see *Figure 4*. The euglenid flagellum is the composite structure consisting of Ax and PFR attached together. We suppose that the Ax is the only active component of the flagellum, whereas the PFR is purely passive. Our mechanical model builds on the definition of the total internal energy of the flagellum

$$\mathcal{W} = \mathcal{W}^a_{pas} + \mathcal{W}^a_{act} + \mathcal{W}^p \tag{1}$$

which is given by the sum of three terms: the passive (elastic) internal energy $\mathcal{W}^a_{pas}$ of the Ax, the active internal energy $\mathcal{W}^a_{act}$ of the Ax (generated by dynein action), and the (passive, elastic) internal energy $\mathcal{W}^p$ of the PFR. The passive internal energy of the Ax is given by

$$\mathcal{W}^a_{pas} = \frac{1}{2} \int_0^L B^a \left( U_1(s)^2 + U_2(s)^2 \right) + C^a U_3(s)^2 \, ds, \tag{2}$$

where $U_1$ and $U_2$ are the bending strains of the Ax, $U_3$ is the twist, $B^a$ and $C^a$ are the bending and twist moduli (respectively), and $L$ is the total length of the Ax centerline $\mathbf{r}^a$. Bending strains and twist

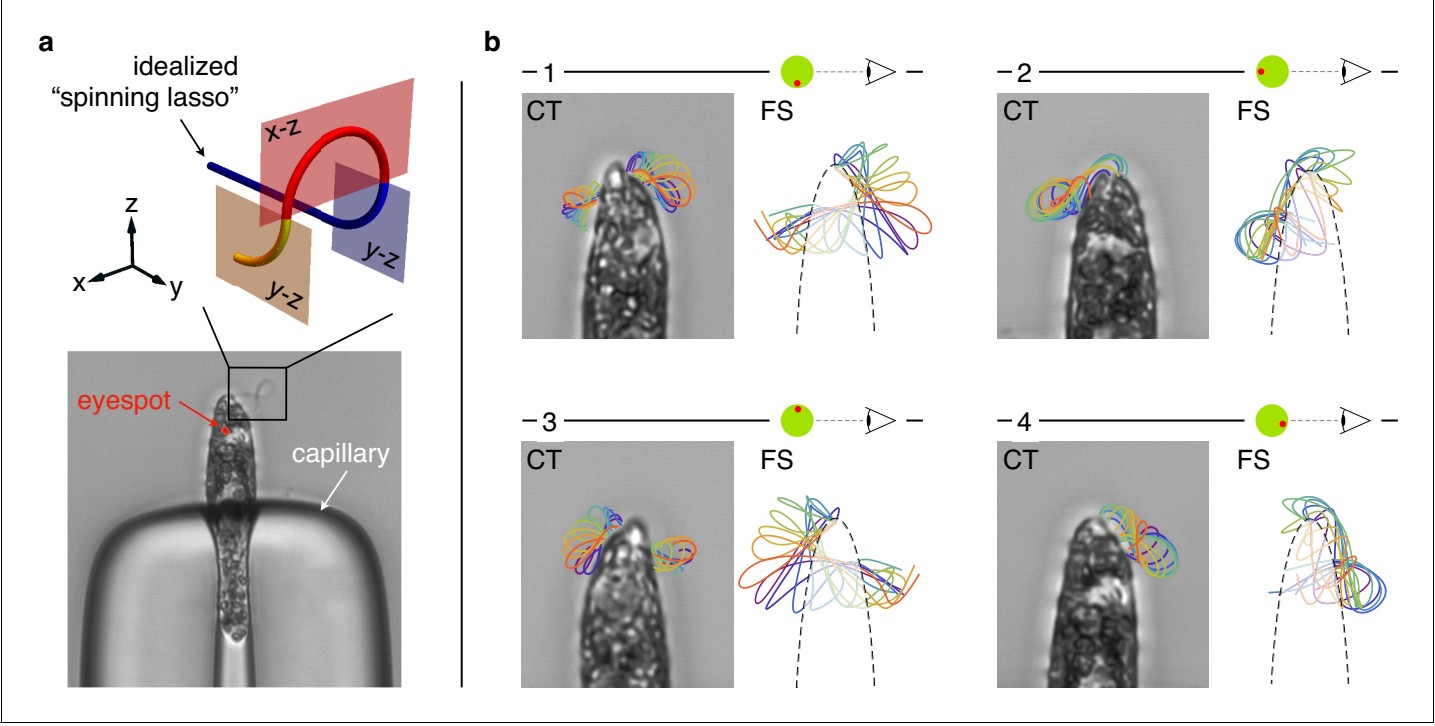

**Figure 3.** Flagellar beat of capillary trapped specimens. (**a**) A specimen of *Euglena gracilis* trapped at the tip of a capillary (bottom). The typical outline of its beating flagellum is that of a looping curve, which is consistent with the outline of a simple curve with two concentrated torsional peaks of alternate sign along the length of the curve, that is, a torsion dipole (top). (**b**) Close-up images of the same specimen of capillary-trapped (CT) *E. gracilis* as seen from different viewpoints, upon successive ~90° turns of the capillary tube. The body orientation with respect to the objective is estimated from the anatomy of the cell, and in particular from the position of the eyespot (a visible light-sensing organelle present on the cell surface). Microscopy images are decorated with the tracked outlines of the flagellum in different phases (same color coding as in *Figure 2*). The outlines (2d projections) of the 3d reconstructed flagellar beat of freely swimming (FS) specimens are shown for comparison.

depend on the arc length $s$ of the centerline, and

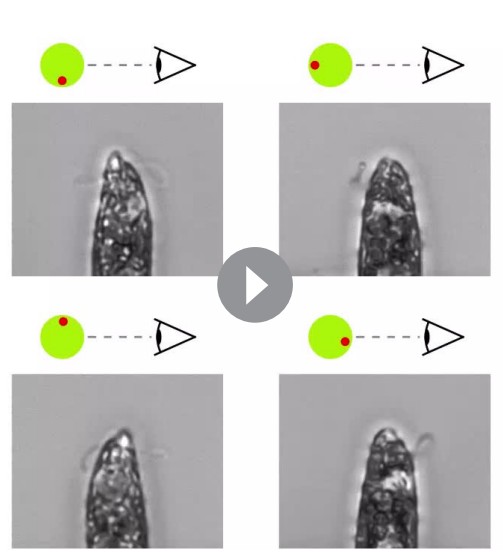

**Video 1.** Four views of a capillary-trapped specimen of *E. gracilis* recorded during periodic flagellar beating. https://elifesciences.org/articles/58610#video1

they are defined as follows. We associate to the curve $\mathbf{r}^a$ an orthonormal frame $\mathbf{d}_i(s)$, with $i = 1, 2, 3$, which determines the orientation of the orthogonal sections of the Ax (enclosed by light blue circles in *Figure 4*). The unit vectors $\mathbf{d}_1(s)$ and $\mathbf{d}_2(s)$ define the plane of the orthogonal section at $s$. The unit vector $\mathbf{d}_1(s)$ lies on the line that connects the center of the Ax to MT 2, the center of the bonding links complex, see *Figure 1*. The unit vector $\mathbf{d}_3(s) = \partial_s \mathbf{r}^a(s)$ lies perpendicular to the orthogonal section. Bending strains and twist are then given by

$$U_1 = \partial_s \mathbf{d}_2 \cdot \mathbf{d}_3 , \quad U_2 = \partial_s \mathbf{d}_3 \cdot \mathbf{d}_1 ,$$
$$\text{and} \quad U_3 = \partial_s \mathbf{d}_1 \cdot \mathbf{d}_2 . \tag{3}$$

Thus, $U_1$ and $U_2$ measure the bending of the Ax on the local planes $\mathbf{d}_2$-$\mathbf{d}_3$ and $\mathbf{d}_3$-$\mathbf{d}_1$, respectively, while the twist $U_3$ is given by the rotation rate of the orthonormal frame around the tangent $\mathbf{d}_3$ to the centerline.

We remark that the right-hand side of *Equation 2* is formally identical to a classical expression arising in Kirchhoff's theory for elastic rods

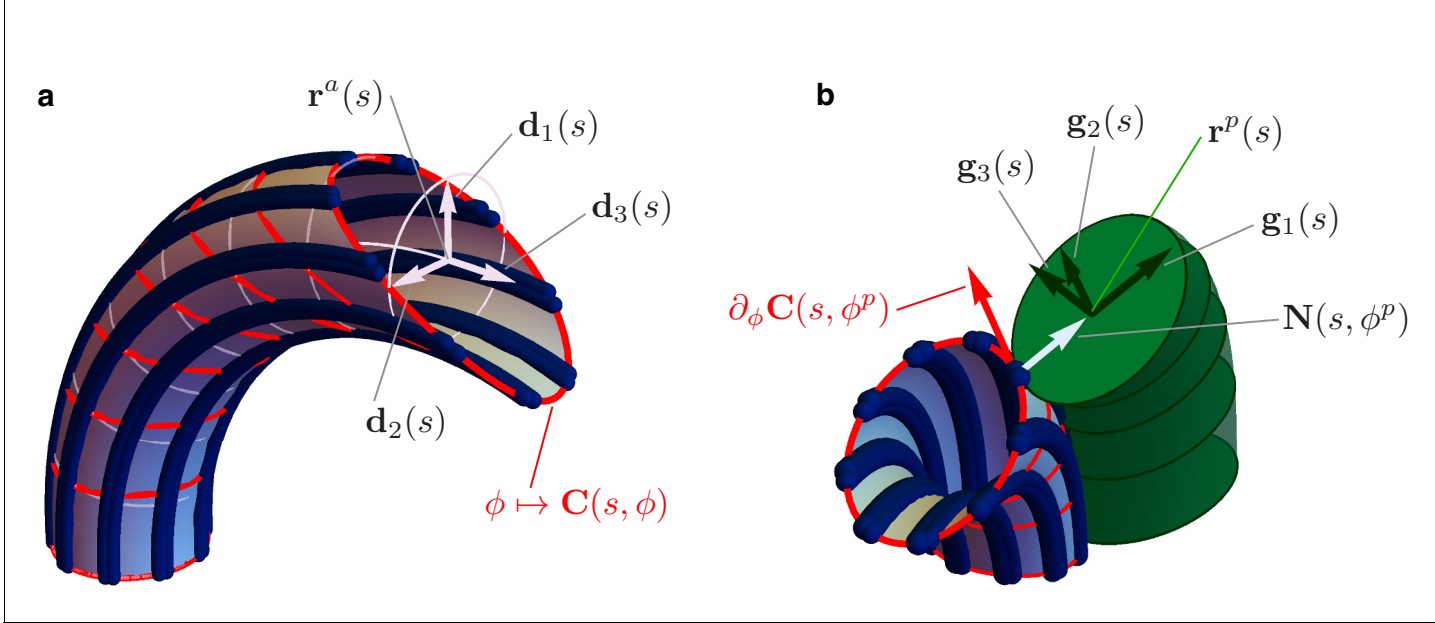

**Figure 4.** Details of the mechanical model. (a) Geometry of the Ax. MTs lie on a tubular surface $\mathbf{C}(s,\phi)$ parametrized by generalized polar coordinates $s$ and $\phi$, where $s$ is the arc length of the axonemal centerline $\mathbf{r}^a$. The unit vectors $\mathbf{d}_1(s)$ and $\mathbf{d}_2(s)$ lie on the orthogonal cross sections of the Ax (light blue circles). The material sections of the Ax are given by the curves $\phi\mapsto\mathbf{C}(s,\phi)$ (red), which connect points of neighbouring axonemal MTs corresponding to the same arc length $s$. Bend deformations of the axoneme are generated by the shear (collective sliding) of MTs. The shear is quantified by the angle between the orthogonal sections and the material sections of the Ax. (b) Geometry of the euglenid flagellum, detail of the Ax-PFR attachment. The unit vectors $\mathbf{g}_1(s)$ and $\mathbf{g}_2(s)$ generate the plane of the PFR's cross sections. The vector $\mathbf{g}_1(s)$ is parallel to the outer unit normal to the axonemal surface $\mathbf{N}(s,\phi^p)$, while $\mathbf{g}_2(s)$ is parallel to the tangent vector to the material section $\partial_\phi\mathbf{C}(s,\phi^p)$.

(*Goriely, 2017*). Our rod is however non-standard because it consists of a hollow tubular structure arising as the envelope of nine individual MTs. In Appendix 1, we model each of the MTs as a standard rod with (full cross-section and) bending and twisting moduli $B^m$ and $C^m$. We then show that the geometry and deformations of the Ax (centerline $\mathbf{r}^a$, frame vectors $\mathbf{d}_1$, $\mathbf{d}_2$, and $\mathbf{d}_3$, bending strains and twist) determine the geometry and deformations of the individual MTs. By summing the elastic contributions of individual MTs, we obtain *Equation 2* as the elastic energy of the assembly, with $B^a = 9B^m$ and $C^a = 9C^m$.

The active internal energy of the Ax is defined as minus the total mechanical work of the dyneins

$$\mathcal{W}^a_{act} = -\int_0^L \left(H_1(s)\gamma_1(s) + H_2(s)\gamma_2(s)\right) ds - \left(\widehat{H}_1\gamma_1(L) + \widehat{H}_2\gamma_2(L)\right), \tag{4}$$

where $\gamma_1$ and $\gamma_2$ are the two variables that quantify the shear (i.e., collective sliding) of MTs, while $H_1$ and $H_2$ are the corresponding shear forces exerted by molecular motors. Following *Sartori et al., 2016b* we also allow for singular shear forces, $\widehat{H}_1$ and $\widehat{H}_2$, concentrated at the distal end of the Ax. These forces arise naturally, as we remark after *Equation 23* in the 'Results' Section.

The active internal energy can be written in a more natural way at the level of individual MT pairs in terms of the work done by the sliding forces $F_j$ generated by the dyneins cross-bridging MTs $j$ and $j+1$ against their relative sliding displacements $\sigma_j$, for $j=1,2,\ldots,9$. These forces and displacements are defined in detail in Appendix 1, and their work computed in *Equation 35*. The way *Equation 35* gives rise to the equivalent reformulation (*Equation 4*) in terms of global cross-section variables, the forces $H_i$ and shears $\gamma_i$, is also discussed there. Here, we simply notice that the structural constraints of the Ax lead to simplifications on the kinematics. These constraints do not allow MTs to slide by whatever amount, and the sliding of MT pairs $\sigma_j$ are not independent. Rather, there are only two degrees of freedom that determine MTs sliding, which are given by the shear variables $\gamma_1$ and $\gamma_2$. Moreover, again due to the structural constraints of the Ax, the shear variables are coupled to the bending strains (*Equation 3*), as discussed further below. In Appendix 1, we derive the

linear relations between the shear variables and the individual sliding of MT pairs (*Equation 36*). We also compute the relations between the dynein forces $F_j$ acting on each pair of adjacent MTs and the shear forces $H_1$ and $H_2$ (*Equation 37*). The singular shear forces $\widehat{H}_1$ and $\widehat{H}_2$ arise from concentrated sliding forces $\widehat{F}_j$ at the distal end of the Ax in an analogous way.

To explain how the shear variables $\gamma_1$ and $\gamma_2$ are related to the MTs' kinematics, we observe that the MTs centerlines $\mathbf{r}^j$, for $j = 1, 2 \ldots 9$, are given by $\mathbf{r}^j(s) = \mathbf{C}(s, \phi_j)$, where $\phi_j = 2\pi(2 - j)/9$, and

$$\mathbf{C}(s, \phi) \approx \mathbf{r}^a(s) + \rho^a \left( \cos\phi \, \mathbf{d}_1(s) + \sin\phi \, \mathbf{d}_2(s) + (\cos\phi \, \gamma_1(s) + \sin\phi \, \gamma_2(s)) \mathbf{d}_3(s) \right) \tag{5}$$

is the parametrization of the cylindrical surface of the Ax ($\rho^a$ is the Ax radius) in terms of the centerline arc length $s$ and the angle $\phi$. A special axonemal deformation with $\gamma_2 = 0$ is shown in *Figure 4*. In this case, the Ax is bent into a circular arc, and the centerline $\mathbf{r}^a$ lies on the plane generated by the unit vectors $\mathbf{d}_1$ and $\mathbf{d}_3$. The shear variable $\gamma_1(s) \neq 0$ has here a simple geometrical interpretation. For each fixed $s$ the curve $\phi \mapsto \mathbf{C}(s, \phi)$ describes what we call the 'material' section of the Ax at $s$ (red curves in *Figure 4*). The material section is a planar ellipse centered in $\mathbf{r}^a(s)$ which connects points of neighboring MTs' corresponding to the same arc length. *Equation 5* says that $\gamma_1(s)$ is the tangent of the angle at which the material sections at $s$ intersect the orthogonal sections at $s$.

As mentioned above, the kinematic constraints of the Ax couple the shear variables with bending strains. We have

$$\gamma_1(s) = \int_0^s U_2 \quad \text{and} \quad \gamma_2(s) = -\int_0^s U_1 \,. \tag{6}$$

The above formulas (whose detailed derivation is given in Appendix 1) underlie the essential mechanism of axonemal motility: collective sliding of MTs generates bending of the whole Ax. We point out here that there is no coupling between the shear variables $\gamma_1, \gamma_2$ and the twist $U_3$, a fact that will have consequences in the remainder.

The special axonemal deformation in *Figure 4* shows the case in which $U_1(s) = 0$ and $U_2(s) = K$, so that the Ax is bent into a circular arc of radius $1/K$. While $\gamma_2(s) = 0$, the shear variable $\gamma_1(s) = Ks$ increases linearly with $s$. Material and orthogonal sections coincide at the base (the basal body impose no shear at $s = 0$) and the angle between them grows as we move along the centerline towards the distal end of the Ax. In order for the Ax to bend, MTs from one side of the Ax must be driven toward the distal end while the others must be driven toward the proximal end.

We remark here that *Equation 4* defines the most general active internal energy generated by molecular motors, and we do not assume at this stage any specific (spatial) organization of dynein forces. We will introduce specific shear forces later in the 'Results' Section.

The PFR is modelled as an elastic cylinder with circular cross sections of radius $\rho^p$ and rest length $L$. We assume that the PFR can stretch and shear. The total internal energy of the PFR is given by

$$\mathcal{W}^p = \frac{1}{2} \int_0^L D^p (V_1(s)^2 + V_2(s)^2) + E^p V_3(s)^2 \, ds \tag{7}$$

where $V_1$ and $V_2$ are the shear strains, $V_3$ is the stretch, $D^p$ and $E^p$ are the shear and stretching moduli, respectively. We are neglecting here the PFR's bending and twisting stiffness. Classical estimations on homogeneous elastic rods, see, for example *Goriely, 2017*, show that bending and twist moduli scale with the fourth power of the cross section radius, whereas shear and stretching moduli scale with the second power and hence they are dominant for small radii. We assume that dynein forces are strong enough to induce shear in the PFR, thus PFR's bending and twist contributions to the energy of the flagellum become negligible. We are also neglecting Poisson effects by treating the PFR cross-sections as rigid.

The PFR shear strains and stretch are defined as follows. The cross-sections centers of the PFR lie on the curve $\mathbf{r}^d$, and their orientations are given by the orthonormal frame $\mathbf{g}_i(s)$, with $i = 1, 2, 3$. The unit vectors $\mathbf{g}_1(s)$ and $\mathbf{g}_2(s)$ determine the cross section plane centered at $\mathbf{r}^p(s)$, while the unit vector $\mathbf{g}_3(s)$ is orthogonal to it. The curve $\mathbf{r}^p$ is not parametrized by arc length and $\mathbf{g}_3$ is not in general aligned with the tangent to $\mathbf{r}^p$. Shear strains and stretch are given by the formulas

$$V_1 = \partial_s \mathbf{r}^p \cdot \mathbf{g}_1, \quad V_2 = \partial_s \mathbf{r}^p \cdot \mathbf{g}_2, \quad \text{and} \quad V_3 = \|\partial_s \mathbf{r}^p\| - 1 \,. \tag{8}$$

The shear strains thus depend on the orientation of the cross sections with respect to the centerline (tangent), while the stretch measures the elongation of the centerline.

The PFR-Ax attachment couples the kinematics of the two substructures, see *Figure 4*. In the remainder, we formalize the attachment constraint and we show how the PFR's shear strains and stretch (*Equation 8*), and thus the flagellar energy (*Equation 1*), are completely determined by the Ax kinematic variables.

For each $s$, the PFR cross-section centered at $\mathbf{r}^p(s)$ is in contact with the Ax surface at the point $\mathbf{C}(s, \phi^p)$ for a fixed angle coordinate $\phi^p$, see *Figure 1* and *Melkonian et al., 1982*. The PFR centerline is given by

$$\mathbf{r}^p(s) = \mathbf{C}(s, \phi^p) + \rho^p \mathbf{N}(s, \phi^p),\qquad(9)$$

where $\mathbf{N}(s, \phi^p) \approx \mathbf{d}_1(s) \cos \phi^p + \mathbf{d}_2(s) \sin \phi^p$ is the outer unit normal to the axonemal surface at $\mathbf{C}(s, \phi^p)$. The normal vector $\mathbf{N}(s, \phi^p)$ lies on the plane of the PFR cross-section centered at $\mathbf{r}^p(s)$. Indeed, we have $\mathbf{g}_1(s) = \mathbf{N}(s, \phi^p)$ for the first unit vector of the PFR orthonormal frame. Only one more degree of freedom remains, namely $\mathbf{g}_2(s)$, which must be orthogonal to $\mathbf{N}(s, \phi^p)$, to fully characterize the orientations of the PFR cross-sections. Here is where the bonding links attachments are introduced in the model. The bonding links of the PFR cross-section centered at $\mathbf{r}^p(s)$ are attached to three adjacent MTs at the same MTs' arc length $s$. The individual attachments are therefore located on the material section of the Ax at $s$. Given this, $\mathbf{g}_2(s)$ is imposed to be parallel to $\partial_\phi \mathbf{C}(s, \phi^p)$, the tangent vector to the material section of the Ax at the point of contact $\mathbf{C}(s, \phi^p)$, see *Figure 4*. This condition critically couples MTs' shear to the orientations of the PFR cross-sections, as further demonstrated below.

To summarize, we have the following formulas for the PFR orthonormal frame vectors

$$\mathbf{g}_1(s) = \mathbf{N}(s, \phi^p), \quad \mathbf{g}_2(s) = \partial_\phi \mathbf{C}(s, \phi^p) / \|\partial_\phi \mathbf{C}(s, \phi^p)\|, \quad \text{and} \quad \mathbf{g}_3(s) = \mathbf{g}_1(s) \times \mathbf{g}_2(s).\qquad(10)$$

By replacing the expressions in *Equations 9-10* in *Equation 8*, we obtain formulas for the shear strains and stretch of the PFR in terms of the Ax kinematic parameters. The shear strain $V_1$ and the stretch $V_3$ are found to be of order $\rho^p \sim \rho^a$ (see Appendix 1 for detailed calculations). Since $\rho^p$ is small compared to the length scale $L$ of both PFR and Ax, we neglect these quantities. The only non-negligible contribution to the PFR energy is thus given by the shear strain $V_2$. After linearization, we have $V_2 \approx -\sin \phi^p \gamma_1 + \cos \phi^p \gamma_2$. The PFR energy in terms of Ax kinematic parameters is then given by

$$\mathcal{W}^p \approx \frac{1}{2} \int_0^L D^p \left( -\sin \phi^p \gamma_1(s) + \cos \phi^p \gamma_2(s) \right)^2 ds.\qquad(11)$$

The shear of axonemal MTs determines the orientation of the PFR cross-sections. In *Figure 5* (middle pictures), we show an example of this kinematic interplay. The Ax is again bent in an arc of a circle on the plane $\mathbf{d}_1 - \mathbf{d}_3$, with $U_1(s) = 0$, $U_2(s) = K$, $\gamma_1(s) = Ks$, and $\gamma_2(s) = 0$. PFR and Ax centerlines run parallel to each other, indeed from *Equation 9* we have that $\partial_s \mathbf{r}^a \approx \partial_s \mathbf{r}^p$ for every deformation. The linking bonds impose a rotation of the cross sections of the PFR as we progress from the proximal to the distal end of the flagellum, generating shear strain $V_2(s) = -\sin \phi^p \gamma_1(s) = -\sin \phi^p Ks$ on the PFR. This mechanical interplay leads to non-planarity of the euglenid flagellar beat. This mechanism is controlled by the offset between the PFR-Ax joining line and the local spontaneous bending plane of the Ax, as further discussed in the 'Results' Section.

## Equilibria

Under generic (steady) dynein actuation, that is, given $H_1$ and $H_2$ (not time-dependent), and in the absence of external forces, the flagellum deforms to its equilibrium configuration $\delta \mathcal{W} = 0$. Bending strains and twist at equilibrium solve the equations

$$B^a \partial_s \mathbf{U} - \mathbf{H}^\perp - D^p \mathbf{e}_p \otimes \mathbf{e}_p \int_0^s \mathbf{U} = \mathbf{0} \quad \text{and} \quad C^a \partial_s U_3 = 0,\qquad(12)$$

where $\mathbf{U} = (U_1, U_2)$ is the bending vector, $\mathbf{e}_p = (\cos \phi^p, \sin \phi^p)$, and $\mathbf{H}^\perp = (-H_2, H_1)$. We use the symbol $\mathbf{a} \otimes \mathbf{b}$ to denote the matrix with components $(\mathbf{a} \otimes \mathbf{b})_{ij} = a_i b_j$. The field *Equation 12* is complemented by the boundary conditions

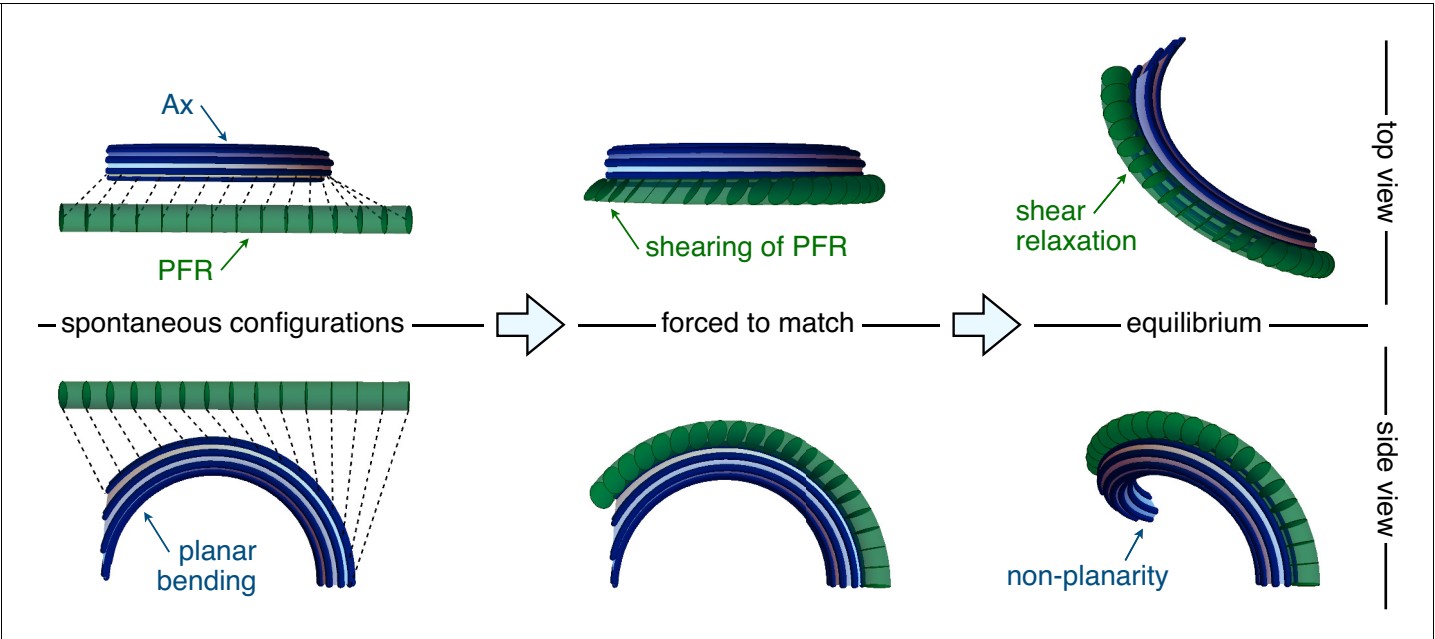

**Figure 5.** Flagellar non-planarity arising from structural incompatibility. The Ax-PFR mechanical interplay is explained in a three-steps argument (left to right). Consider first the two separated structures in their spontaneous configurations (left). The Ax is bent into a planar arc while the PFR is straight. Then, the PFR is forced to match to the Ax, while the latter is kept in its spontaneous configuration (middle). The attachment constraint induces shear strains in the PFR, such that the composite system cannot be in mechanical equilibrium without external forcing. When the composite system is released (right), it reaches equilibrium by the relaxation of the PFR shear, which induces additional distortion of the Ax. At equilibrium, an optimal energy compromise is reached, which is characterized by an emergent non-planarity.

$$B^a \mathbf{U}(L) + \widehat{\mathbf{H}}^\perp = \mathbf{0} \quad \text{and} \quad U_3(L) = 0, \tag{13}$$

where $\widehat{\mathbf{H}}^\perp = (-\widehat{H}_2, \widehat{H}_1)$. **Equations 12, 13** can be interpreted as the torque balance equations of the Ax. The derivative of the (elastic) bending moment and the internal shear stresses balance the torque per unit length exerted by the PFR on the Ax, which is given by the $D^p$-dependent term appearing in the first equation. The torque depends on the integral of the bending vector, making the balance equations non-standard (integrodifferential instead of differential). This dependency is due to the fact that the torque arises from the shear deformations of the PFR, which are induced by the shear of axonemal MTs, which is, in turn, related to axonemal bending strains via the integral relations (**Equation 6**). The torque exerted by the PFR on the Ax is sensitive to the direction given by the unit vector $\mathbf{e}_p$, hence it depends on the angle $\phi^p$ between the Ax-PFR joining line and the unit vector $\mathbf{d}_1$.

## Hydrodynamics

We consider here our mechanical model in the presence of external forces. For simplicity, we ignore the possible forces exerted by the cell surface on the non-emergent portion of the flagellum, located inside the reservoir of the cell. Indeed, we suppose that the flagellum is immotile and straight in the region inside the reservoir. We can assume, therefore, that our model effectively describes the flagellum from its emergence point outwards. The only forces the flagellum is subject to come from fluid interaction, which we assume to act all along its length. We consider the extended functional

$$\mathcal{L} = \mathcal{W} + \int_0^L \mathbf{\Lambda} \cdot (\partial_s \mathbf{r}^a - \mathbf{d}_3) \tag{14}$$

where $\mathbf{\Lambda}$ is the Lagrange multiplier vector enforcing the constraint $\partial_s \mathbf{r}^a = \mathbf{d}_3$. We treat the fluid-flagellum interaction in the local drag approximation of Resistive Force Theory, see for example *Wolgemuth et al., 2000*. In this approximation, viscous forces and torques depend locally on the

translational and rotational velocity of the flagellum, represented here for simplicity by the translational and rotational velocity of the Ax. The external viscous forces $\mathbf{F}$ and torques $\mathbf{G}$ (per unit length) acting on the flagellum are given by

$$\mathbf{F} = -\mu_\perp(\mathbf{Id} - \mathbf{d}_3 \otimes \mathbf{d}_3)\partial_t\mathbf{r}^a - \mu_{||}\mathbf{d}_3 \otimes \mathbf{d}_3\partial_t\mathbf{r}^a \quad \text{and} \quad \mathbf{G} = -\mu_r(\partial_t\mathbf{d}_1 \cdot \mathbf{d}_2)\mathbf{d}_3 \,, \tag{15}$$

where $\mu_\perp$, $\mu_{||}$, and $\mu_r$ are the normal, parallel, and rotational drag coefficient (respectively), and $\mathbf{Id}$ is the identity tensor. The principle of virtual work imposes

$$\delta\mathcal{L} = \int_0^L \mathbf{F} \cdot \delta\mathbf{r}^a + \mathbf{G} \cdot \delta\boldsymbol{\theta} \tag{16}$$

for every variation $\delta\mathbf{r}^a$ and $\delta\boldsymbol{\theta} = \delta\theta_1\mathbf{d}_1 + \delta\theta_2\mathbf{d}_2 + \delta\theta_3\mathbf{d}_3$, where $\delta\theta_1 = (\delta\mathbf{d}_2 \cdot \mathbf{d}_3)$, $\delta\theta_2 = (\delta\mathbf{d}_3 \cdot \mathbf{d}_1)$, and $\delta\theta_3 = (\delta\mathbf{d}_1 \cdot \mathbf{d}_2)$. Linearizing the force balance equations derived from *Equation 16* we obtain the following equations for bending strains and twist

$$\mu_\perp\partial_t\mathbf{U} = -B^a\partial_s^4\mathbf{U} + \partial_s^3\mathbf{H}^\perp + D^p\mathbf{e}_p \otimes \mathbf{e}_p\partial_s^2\mathbf{U} \tag{17}$$

$$\text{and} \quad \mu_r\partial_t U_3 = C^a\partial_s^2 U_3 \,, \tag{18}$$

which are decoupled from the extra unknown $\boldsymbol{\Lambda}$. *Equations 17 and 18* are complemented by the boundary conditions

$$B^a\mathbf{U}|_{s=L} + \widehat{\mathbf{H}}^\perp = \mathbf{0}, \quad \left(B^a\partial_s\mathbf{U} - \mathbf{H}^\perp - D^p\mathbf{e}_p \otimes \mathbf{e}_p\int_0^s\mathbf{U}\right)|_{s=L} = \mathbf{0}, \quad U_3|_{s=L} = 0, \quad \partial_s U_3|_{s=L} = 0, \tag{19}$$

$$\left(B^a\partial_s^2\mathbf{U} - \partial_s\mathbf{H}^\perp - D^p\mathbf{e}_p \otimes \mathbf{e}_p\mathbf{U}\right)|_{s=0} = \mathbf{0}, \quad \text{and} \quad \left(B^a\partial_s^3\mathbf{U} - \partial_s^2\mathbf{H}^\perp - D^p\mathbf{e}_p \otimes \mathbf{e}_p\partial_s\mathbf{U}\right)|_{s=0} = \mathbf{0}. \tag{20}$$

The details of the derivation of *Equations 17–20* are provided in Appendix 2.

Once we solve for $U_1$, $U_2$, and $U_3$ either the equilibrium *Equations 12, 13* or the dynamic *Equations 17–20*, the shape of the flagellum can be recovered. In particular, we obtain the orthonormal frame $\mathbf{d}_i$ with $i = 1, 2, 3$ by solving *Equation 53*, while the centerline of the Ax is recovered by integrating $\partial_s\mathbf{r}^a = \mathbf{d}_3$.

## Results

We analyze the geometry of the centerline $\mathbf{r}^a$ which, due to the slenderness of the flagellar structure, is a close proxy for the shape of the flagellum.

In general, the shape of a curve is determined by its curvature $\kappa$ and torsion $\tau$. Since $\mathbf{r}^a$ is parametrized by arc length, the two quantities are given by the formulas $\partial_s\mathbf{t} = \kappa\mathbf{n}$ and $\partial_s\mathbf{b} = -\tau\mathbf{n}$, where $\mathbf{t} = \partial_s\mathbf{r}^a$, $\mathbf{n} = \partial_s\mathbf{t}/|\partial_s\mathbf{t}|$, and $\mathbf{b} = \mathbf{t} \times \mathbf{n}$ are the tangent, normal, and binormal vector to the curve $\mathbf{r}^a$, respectively. Given $\kappa$ and $\tau$, $\mathbf{r}^a$ is uniquely determined up to rigid motions.

From the previous definitions and from *Equation 3* we obtain the relations between curvature, torsion, bending, and twist. In compact form these relations are given by

$$U_1 + iU_2 = \kappa e^{i\psi} \quad \text{and} \quad \tau = \partial_s\psi + U_3 \,, \tag{21}$$

which hold for $\mathbf{U} \neq 0$. In *Equation 21* we introduced the angle $\psi$ that the bending vector $\mathbf{U} = (U_1, U_2)$ forms with the line $U_2 = 0$, see *Figure 6*. Now, at equilibrium (*Equation 12*) we have

$$U_3 = 0 \,, \tag{22}$$

under any dynein actuation. In other words, axonemal deformations are twistless. This is, fundamentally, a consequence of the fact that shear of axonemal MTs and twist are uncoupled (*Equation 6*). The torsion of the centerline $\mathbf{r}^a$ is our main focus, since we are interested in emergent non-planarity. Combining *Equation 21* and *Equation 22* we have that torsion can arise only from the rotation rate $\partial_s\psi$ of the bending vector $\mathbf{U}$ along the length of the flagellum. This last observation will be important in the following.

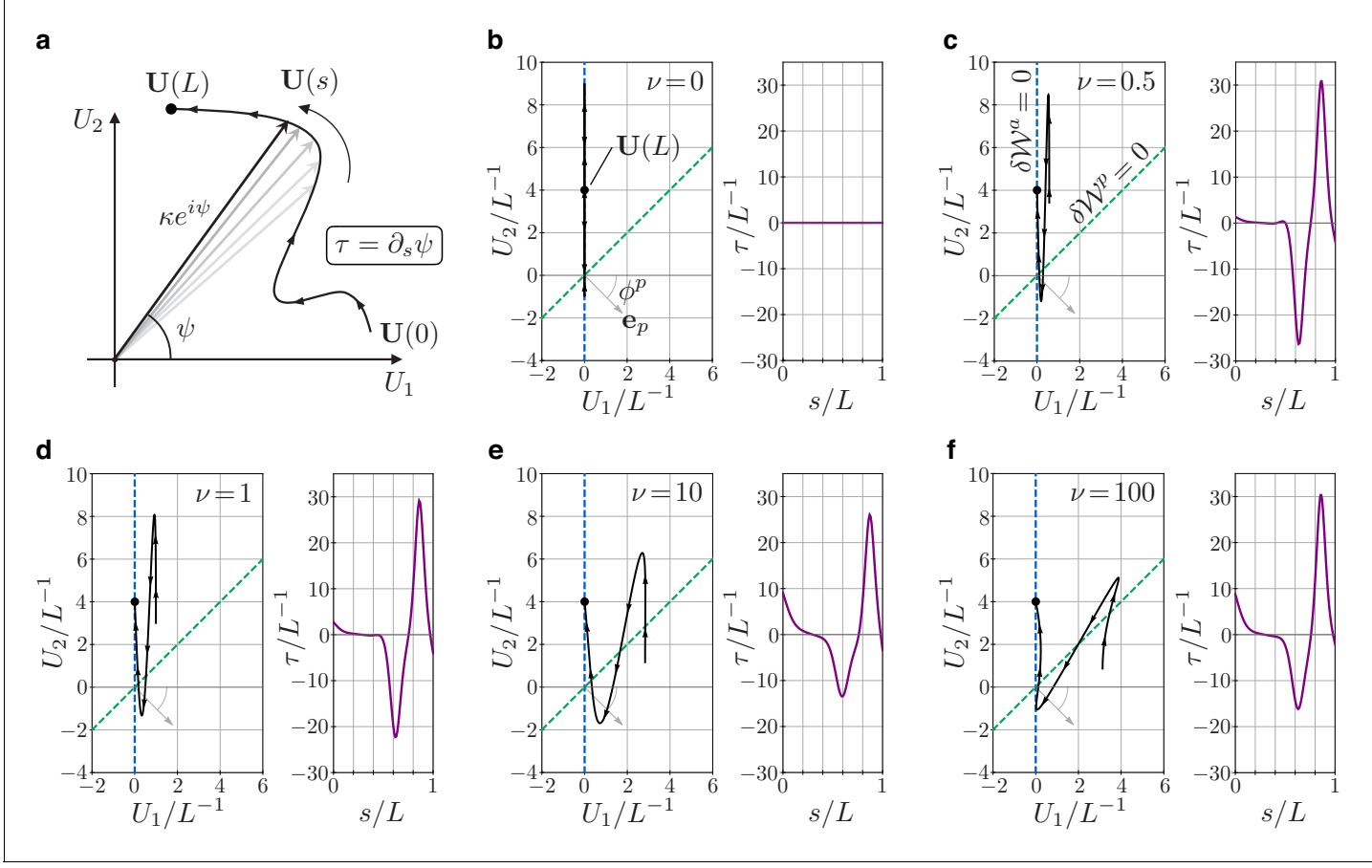

**Figure 6.** Geometry and mechanics of non-planar flagellar shapes. (a) The bending vector $\mathbf{U}(s) = (U_1(s), U_2(s))$ traces a curve on the plane of the bending parameters $U_1$ and $U_2$. The norm of the bending vector determines the curvature $\kappa(s) = |\mathbf{U}(s)|$ of the flagellum. The rate of change of the angle $\psi(s)$ determines the torsion $\tau = \partial_s \psi$. (b–f) Bending vectors' traces of flagellar equilibrium configurations under the same (steady) dynein actuation, but different values of the material parameter $\nu = D^p/(B^a L^{-2})$. Equilibria are minimizer of the energy $\mathcal{W} = \mathcal{W}^a + \mathcal{W}^p$. For small values of $\nu$, the Ax component of the energy $\mathcal{W}^a$ dominates. In this case, $\mathbf{U}$ is close to the target bending vector $(0, U_2^*)$ where $U_2^*(s) = A_0 + A_1 \sin(2\pi s/L)$. For large values of $\nu$ the PFR component of the energy $\mathcal{W}^p$ dominates, and equilibria are dragged closer to the line orthogonal to the vector $\mathbf{e}_p$ (dashed green). The bending vector undergoes rotations which result in torsional peaks of alternating sign.

The online version of this article includes the following source code for figure 6:

**Source code 1.** Equilibrium equations solver.

## Dyneins' actuation induced by sliding inhibition

Under the assumptions *Equation 22* and *Equation 24*, the flagellar energy (*Equation 1*) can be rewritten as

$$\mathcal{W} = \frac{1}{2}\int_0^L B^a \left\| \mathbf{U} - \begin{pmatrix} U_1^* \\ U_2^* \end{pmatrix} \right\|^2 + D^p \left( \int_0^s \mathbf{e}_p \cdot \mathbf{U} \right)^2 - B^a (U_1^{*2} + U_2^{*2}),$$

$$\text{where} \quad U_1^*(s) = \left( \widehat{H}_2 + \int_s^L H_2 \right)/B^a \quad \text{and} \quad U_2^*(s) = -\left( \widehat{H}_1 + \int_s^L H_1 \right)/B^a$$

(23)

are the target bending strains generated by the dynein forces. The use of this terminology is clear from *Equation 23*. The effect of dynein actuation at equilibrium (when the energy is minimized) is to bring the bending strains $U_1$ and $U_2$ as close as possible to $U_1^*$ and $U_2^*$, respectively. The emerging bending strains and the target bending strains might not match due to the interference by the PFR component of the energy ($D^p \neq 0$). From the formulas for the target bending strains in *Equation 23,* we can infer the importance of the concentrated shear forces $\widehat{H}_1$ and $\widehat{H}_2$. Without these forces, admissible spontaneous configurations of the Ax would be ruled out. If the concentrated shear

forces are null, for example, the Ax cannot spontaneously bend into a circular arc. Indeed, for a circular arc of radius $1/K$ on the plane $\mathbf{d}_1$-$\mathbf{d}_3$ we must have $U_2^* = 0$ and $U_2^* = K$. In this case, from **Equation 23** we have that $-H_1/B^a = \partial_s U_2^* = 0$, which implies $H_1 = 0$ and $\widehat{H}_1 = -B^a K$, so the concentrated forces must be non null.

Our working hypothesis is that MTs' sliding stretches the Ax-PFR bonding links, which, in turn, inhibits sliding of MTs' 1, 2, and 3, and triggers a dynein organization (via mechanical feedback) similar to the one present in *Chlamydomonas*. We take this feedback-based self-organization process as a given, and we consider a force pattern that produces local spontaneous bending on the plane $\mathbf{d}_1(s)$-$\mathbf{d}_3(s)$, as shown in **Figure 1**. This is equivalent to require that $U_1^* = 0$, which leads to the following condition on the shear forces

$$H_2 = \widehat{H}_2 = 0. \tag{24}$$

## Emergence of non-planarity

We consider here the equilibrium **Equation 12** under the hypothesis (**Equation 24**). We look at the equilibrium configurations for every possible value of the angle $\phi^p$ between the Ax-PFR joining line and the spontaneous bending plane of the Ax, even though the value of actual interest for *E. gracilis* is $\phi^p \approx -2\pi/9$. We can prove analytically the following statement: *if the Ax-PFR joining line is neither parallel nor orthogonal to the spontaneous bending plane of the Ax, then the emergent flagellar shapes are non-planar.*

Indeed, suppose $H_1 \neq 0$. From **Equation 21** and **Equation 22** it follows that the shape of the flagellum is planar ($\tau = 0$) if and only if the angle $\psi$ of the bending vector $\mathbf{U}(s) = (U_1(s), U_2(s))$ is constant. The bending vector must therefore be confined on a line for every $s$. In this case there must be two constants $c_1$ and $c_2$ such that $U_1(s) = c_1 U(s)$ and $U_2(s) = c_2 U(s)$ for some scalar function $U$. Now, if a planar $\mathbf{U}$ is a solution of **Equation 12**, we must have

$$c_1 B^a \partial_s U - D^p \cos\phi^p (c_1 \cos\phi^p + c_2 \sin\phi^p) \int_0^s U = 0, \tag{25}$$

$$c_2 B^a \partial_s U - D^p \sin\phi^p (c_1 \cos\phi^p + c_2 \sin\phi^p) \int_0^s U = H_1, \tag{26}$$

with $c_1 U(L) = 0$ and $c_2 U(L) = -\widehat{H}_1/B^a$. If $\phi^p \notin \{0, \pi/2, \pi, 3\pi/2\}$ the system of **Equations 25, 26** admits no solution. Indeed, suppose first that $\widehat{H}_1 = 0$. Since $H_1 \neq 0$ we must have $(c_1, c_2) \neq (0,0)$. However, in this case, **Equation 25** admits the unique solution $U = 0$, which is incompatible with **Equation 26**. If $\widehat{H}_1 \neq 0$, then the boundary conditions impose $c_1 = 0$, but in this case **Equation 25** has again $U = 0$ as a unique solution, which is incompatible with both the boundary conditions and with **Equation 26**. Our statement is thus proved.

For $\phi^p \approx -2\pi/9$, the characteristic value for *E. gracilis*, the non-planarity of flagellar shapes is not just possible. It is the only outcome under any non-trivial dynein actuation.

## Structural incompatibility and torsion with alternating sign

Alongside the previous analysis, there is a less technical way to infer the emergence of non-planarity from our model. We look here more closely to the flagellum energy, and we think in terms of structural incompatibility between Ax and PFR, seen as antagonistic elements of the flagellum assembly, see **Figure 5**.

Under the assumptions **Equation 22** and **Equation 24**, the flagellar energy is given by

$$\mathcal{W} = \mathcal{W}^a + \mathcal{W}^p, \quad \text{where}$$

$$\mathcal{W}^a = \frac{1}{2}\int_0^L B^a \left\| \mathbf{U} - \begin{pmatrix} 0 \\ U_2^* \end{pmatrix} \right\|^2 - B^a U_2^{*2} \quad \text{and} \quad \mathcal{W}^p = D^p \left( \int_0^s \mathbf{e}_p \cdot \mathbf{U} \right)^2, \tag{27}$$

with $U_2^*$ as in **Equation 23**. The energy has two components, $\mathcal{W}^a$ that depends on the Ax bending modulus $B^a$, and $\mathcal{W}^p$ that depends on the PFR shear modulus $D^p$. We can vary these material parameters and explore what the resulting minima of $\mathcal{W}$, that is, the equilibrium configurations (**Equations 12-13**), must look like. We consider the nondimensional parameter $\nu = D^p/(B^a L^{-2})$. When

$\nu \ll 1$ the Ax component $\mathcal{W}^a$ of the energy dominates. In this case, at equilibrium, the bending vector has to be close to the target bending vector $\mathbf{U} \approx (0, U_2^*)$. In particular, then, $\mathbf{U}(s)$ will be confined near the line $U_1 = 0$ for every $s$. In the case $\nu \gg 1$ the PFR component $\mathcal{W}^p$ dominates, and the energy is minimized when $\mathbf{U}(s)$ lies close to the line generated by the vector $\mathbf{e}_p^\perp = (-\sin\phi_p, \cos\phi_p)$. Clearly, if the latter line is different from $U_1 = 0$, the two extreme regimes $\nu \ll 1$ and $\nu \gg 1$, each of which favours one of the two individual components, aim at two different equilibrium configurations. In other words, Ax and PFR are structurally incompatible.

When neither of the two energy components dominates, the emergence of non-planarity can be intuitively predicted with the following reasoning. In the intermediate case of $\nu \sim 1$, we expect the equilibrium configurations to be a compromise among the two extreme cases, with the bending vector $\mathbf{U}(s)$ being 'spread out' in the region between the two extreme equilibrium lines. The spreading of the bending vector is aided by the concentrated shear force at the tip, which imposes $\mathbf{U}(L) = (0, U_2^*(L))$ irrespectively of the PFR stiffness. The bending vector is then 'pinned' at $s = L$ on the $U_1 = 0$ line while it gets dragged toward the line generated by $\mathbf{e}_p^\perp$ for large values of $\nu$. Hence the spreading. The bending vector will then span an area and, consequently, undergo rotations. Since torsion is determined by the rotation rate of the bending vector ($\tau = \partial_s \psi$), the resulting flagellar shapes will be non-planar.

*Figure 6* illustrates a critical example in which the previous intuitive reasoning effectively plays out. We consider a target bending of the kind $U_2^*(s) = A_0 + A_1 \sin(2\pi s/L)$, a fair idealization of the asymmetric shapes of a *Chlamydomonas*-like flagellar beat (*Geyer et al., 2016*). We take $\phi^p = -\pi/4$ (larger than the *E. gracilis* value, to obtain clearer graphs). For $\nu = 0$, the bending vector lies inside the $U_1 = 0$ line, and its amplitude oscillates. For positive values of $\nu$, when the PFR stiffness is 'turned on', the oscillating bending vector is extruded from the $U_1 = 0$ line. For large values of $\nu$ it gets closer and closer to the line generated by $\mathbf{e}_p^\perp$. The bending vector spans an area and, following the oscillations, it rotates clock-wise and anti-clock-wise generating an alternation in the torsion sign. This is the geometric signature of the spinning lasso.

In Appendix 3, we relaxed the planar constraint (*Equation 24*), and considered different weakly non-planar spontaneous configurations, by perturbing *Chlamydomonas*-like target bending strains. Each configuration presents a different (non-null) torsional profile when the antagonistic mechanical interaction of the PFR is absent $\nu = 0$. For larger values of $\nu$, each perturbation assumes the typical spinning lasso geometry (torsional peaks of alternate sign), regardless of the native ($\nu = 0$) torsional profile. This shows that the PFR-Ax interactions strongly influence the flagellar shape outcome even when the perfectly planar constraint on the spontaneous bending is relaxed.

## Hydrodynamic simulations and comparison with observations

Our model is able to predict the torsional characteristic of the euglenid flagellum in the static case, and in absence of external forces. We test here the model in the more realistic setting of time-dependent dynein actuation in the presence of hydrodynamic interactions.

We first observe that, as in the static case, the dynamic equations for $\mathbf{U}$ and $U_3$ are decoupled (*Equations 17, 18*), and that dynein forces do not affect twist. We have then twistless kinematics under any actuation also in the dynamic case, at least after a time transient. We can simply assume *Equation 22* for all times, so the torsion of $\mathbf{r}^a$ is still completely determined by the bending vector.

We consider a dynein actuation that generates *Chlamydomonas*-like shapes in a flagellum with no extra-axonemal structures. The shear forces $H_1$ and $\widehat{H}_1$ employed in our simulation are shown in *Figure 7*. The same figure also shows the emergent bending strains of a PFR-free flagellum actuated by said forces, beating in a viscous fluid. The dynamic equations for this system are simply *Equations 17, 18* with $D^p = 0$. The resulting bending strains, which generate a planar beat, resembles the experimentally observed *Chlamydomonas* flagellar curvatures reported in *Qin et al., 2015*.

Finally, *Figure 7* presents the emergent bending strains of the beating euglenid (PFR-bearing) flagellum, together with the corresponding flagellar torsion. The spinning lasso torsional signature is clearly present. Indeed, the fluid-structure interaction does not disrupt the Ax-PFR structural incompatibility, which still generates non-planar shapes with traveling waves of torsional peaks with alternating sign and the typical looping-curve outlines, *Figure 2*. All the details on the methods and parameters employed in the simulations are given in Appendix 4. *Video 2* shows a comparison between the simulated flagellar beat and the experimental observations.

**Figure 7.** Kinematics of the beating euglenid flagellum: comparison between theoretical model and experiments. (**a-b**) Dyneins' shear forces. (**c**) Resulting bending strains and torsion for an Ax actuated by the force pattern (**a–b**), beating in a viscous fluid, and free of extra-axonemal structures. The beat is planar (*Chlamydomonas*-like). (**d**) Resulting bending strains and torsion for an euglenid flagellum (composite structure Ax+PFR) actuated by (**a–b**) and beating in a viscous fluid. The Ax-PFR interaction generates torsional peaks with alternate sign traveling from the proximal to the distal end of the flagellum. (**e**) Resulting shapes for the euglenid flagellum at different instants within a beat, and comparison with experimental observations. The online version of this article includes the following source code for figure 7:

**Source code 1.** Flagellar dynamics solver and visualization tool.

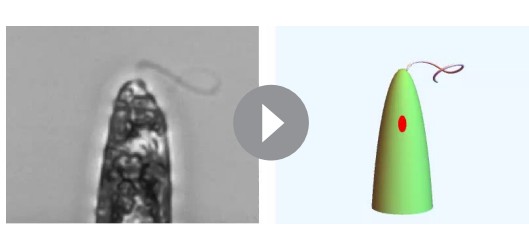

**Video 2.** Comparison between observations of a beating euglenid flagellum and the numerical simulations of our mechanical model.
https://elifesciences.org/articles/58610#video2

## Swimming simulations

The unique flagellar beat of *E. gracilis* is at the base of the distinctive behavior of the organism, producing the typical roto-translational trajectories of swimming cells. This has been demonstrated by swimming simulations using the experimentally measured flagellar shapes in *Rossi et al., 2017* with a Resistive Force Theory approach to model hydrodynamic interactions between the cell and the surrounding fluid. Similar conclusions are reached with a Boundary Element Method for the computation of the fluid

flows induced by the same measured history of the flagellar shapes, see *Giuliani et al., 2021*. We carried out here analogous simulations using the theoretical waveforms produced by our model.

The application of Resistive Force Theory hydrodynamics to the waveforms shown in *Figure 7* produces swimming paths similar to the typical observed trajectories. Swimming cell simulations are reported in *Figure 8*, see also *Video 3*. The cell is propelled by the flagellum, following a generalized right-handed helical trajectory while rotating around its major axis. After each full turn of the helix the cell completes one full body rotation. For more details on the implementation and physics behind swimming simulation see Appendix 5.

## Discussion and outlook

We have shown how the origin of the peculiar shapes of the euglenid flagellum can be explained by the mechanical interplay of two antagonistic flagellar components, the Ax and the PFR. Our conclusions are based mainly on the hypothesis that sliding inhibition by the PFR organizes dynein activity, and localizes the spontaneous bending plane of the Ax as the one that passes from the Ax center through the MTs bonded to the PFR. This is in agreement with the current understanding of the mechanism that generates beat planarity in other PFR-bearing flagellar systems. Non-planarity in *E. gracilis* can arise because of a marked asymmetry in the Ax-bonding links-PFR complex in the euglenid flagellum, which is not found in kinetoplastids such as *Leishmania* (*Gluenz et al., 2010*) or *Trypanosoma* (*Portman and Gull, 2010*).

In the absence of a precise knowledge of the dynein actuation pattern, we tested our mechanical model under shear forces that would, in the absence of extra-axonemal structures, realize a beat similar to those found in model systems like *Chlamydomonas*. We appreciate that the emergent distortion of the Ax, generated by the Ax-PFR interplay, could in principle lead to different actuation patterns, consistently with the hypothesis of dynein actuation via mechanical feedback. Including dynein feedback in the euglenid flagellum model we proposed will require further study, and can potentially open new avenues for the study of ciliary motility in general. While the existence of a mechanical feedback between molecular motors and the flagellar scaffold is fairly accepted, there are several competing theories arguing in favour of different feedback mechanisms. The structural and kinematic peculiarities of the *E. gracilis* beat may provide a challenging new model system to test the relative merits of these alternative theories.

Along with the mechanism that let the euglenid flagellar shapes emerge, it is worth considering how this characteristic flagellar beat is integrated in the overall behavior of the organism. The spinning lasso produces the typical roto-translational motion of *E. gracilis* cells. Cell body rotation is in turn associated with phototaxis. Indeed, rotation allows cells to veer to the light source direction when stimulated, or escape in the opposite direction, when the signal is too strong. Here, the key biochemical mechanism could be the one often found in nature, by which periodic signals generated by lighting and shading associated with body rotations are used for navigation, in the sense that the existence of periodicity implies a lack of proper alignment (*Goldstein, 2015*). It is known that the PFR is directly connected with the light-sensing apparatus (*Rosati et al., 1991*), and might even be contractile (*Piccinni, 1975*). Transient light stimuli have been shown to change flagellar beat

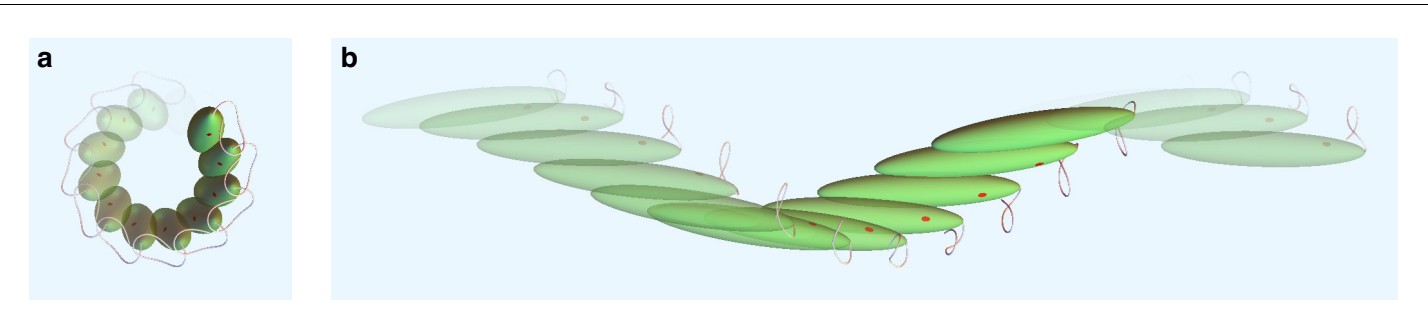

**Figure 8.** Swimming kinematics. (a) Side view and (b) top view of swimming cell simulation resulting from the flagellar beat generated by our model. The dimension of the cell body is not to scale with displacements for visualization purposes.

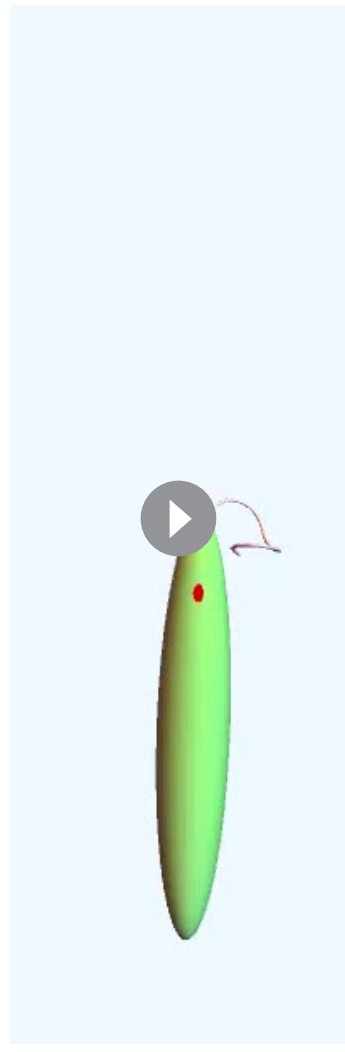

**Video 3.** Simulations of swimming cell kinematics resulting from the flagellar beat generated by our model. https://elifesciences.org/articles/58610#video3

patterns. While the euglenid flagellum consistently present looping outlines (the 2d trace of a torsion dipole), its extension from the cell body changes with changing light intensity signals, resulting in a variety of 'quantized' phototaxis behaviors (*Tsang et al., 2018*). With suitable modifications, our model could provide a starting point to address the (possible) mechanisms of active contraction of the PFR, light perception, and their interplay. Further study on euglenid flagellar motility and phototaxis could lead to a more comprehensive understanding of the biomechanical role of PFR, both in phototaxis and in general.

The very basic question of 'why' euglenids have evolved a structure such as PFR is an interesting challenge, but beyond the scope of the model we have developed in this paper. Our results confirm, however, the interest of euglenids as model systems for responsive unicellular organisms: *E. gracilis*, in particular, is a unicellular organism exhibiting a variety of motility behaviors (flagellar swimming and metaboly, see *Leander et al., 2017*) and capable of responding to a variety of stimuli, from light to confinement, see *Noselli et al., 2019*.

## Materials and methods

Strain SAG 1224-5/27 of *Euglena gracilis* obtained from the SAG Culture Collection of Algae at the University of Göttingen was maintained axenic in liquid culture medium Eg. Cultures were transferred weekly. Cells were kept in an incubator at 15°C at a light:dark cycle of 12 hr under a cold white LED illumination with an irradiance of about $50\,\mu mol\cdot m^{-2}\cdot s^{-1}$.

An Olympus IX 81 inverted microscope with motorized stage was employed in all the experiments. These were performed at the Sensing and Moving Bioinspired Artifacts Laboratory of SISSA. The microscope was equipped with a LCAch 20X Phc objective (NA 0.40) for the imaging of cells trapped at the tip of a glass capillary using transmitted brightfield illumination. The intermediate magnification changer (1.6 X) of the microscope was exploited to achieve higher magnification. Micrographs were recorded at a frame rate of $1,000\,fps$ with a Photron FASTCAM Mini UX100 high-speed digital camera.

Tapered capillaries of circular cross section were obtained from borosilicate glass tubes by employing a micropipette puller and subsequently fire polished. At each trial, observation a glass capillary was filled with a diluted solution of cells and fixed to the microscope stage by means of a custom made, 3d-printed holder. The holder allowed for keeping the capillary in place and rotating it about its axis, so as to image a cell specimen from distinct viewpoints. Cells were immobilized at the tip of the capillary by applying a gentle suction pressure via a syringe connected to the capillary by plastic tubing. Occasionally, large body deformations of the cells were observed, a behavior commonly known as 'metaboly' which *E. gracilis* often manifest under confinement. The observations reported in this paper are restricted to specimens with immotile cell bodies, that is, in absence of metaboly. This choice allows for a clear capture of the flagellar beat. The absence of metaboly also suggests minimal impact on cell behavior in response to capillary entrapment.

## Acknowledgements

We thank A Beran for his assistance with *E. gracilis* samples. This study was supported by the European Research Council through the ERC Advanced Grant 340685-MicroMotility.

## Additional information

### Funding

| Funder | Grant reference number | Author |
|---|---|---|
| European Research Council | 340685-MicroMotility | Antonio DeSimone |
| Ministero dell'Istruzione, dell'Università e della Ricerca | Dipartimenti di Eccellenza 2018-2022 (SISSA - Area Matematica; Scuola Superiore Sant'Anna - Department of Excellence in Robotics and AI) | Giovanni Noselli<br>Antonio DeSimone |

The funders had no role in study design, data collection and interpretation, or the decision to submit the work for publication.

### Author contributions

Giancarlo Cicconofri, Conceptualization, Data curation, Software, Formal analysis, Validation, Investigation, Visualization, Methodology, Writing - original draft, Writing - review and editing; Giovanni Noselli, Resources, Investigation, Visualization, Writing - review and editing; Antonio DeSimone, Conceptualization, Supervision, Funding acquisition, Methodology, Writing - original draft, Project administration, Writing - review and editing

### Author ORCIDs

Giancarlo Cicconofri https://orcid.org/0000-0003-1704-7609
Giovanni Noselli https://orcid.org/0000-0001-8637-713X
Antonio DeSimone https://orcid.org/0000-0002-2632-3057

### Decision letter and Author response

Decision letter https://doi.org/10.7554/eLife.58610.sa1
Author response https://doi.org/10.7554/eLife.58610.sa2

## Additional files

### Supplementary files

• Transparent reporting form

### Data availability

All data generated or analysed during this study are included in the manuscript and supporting files. Source data and code files have been provided for Figures 2, 6, and 7 in the main text, and for Figure 1 in Appendix 5.

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

# Appendix 1

## Model details

The Ax consists of a bundle of inextensible filaments of length $L$ (MTs) lying on a cylindrical surface of radius $\rho^a$. For simplicity, the model ignores the mechanical effects of radial spokes and the central pair. The axonemal surface is parametrized by the generalized cylindrical coordinates $z$ and $\phi$ via the map

$$\boldsymbol{\chi}(z,\phi) = \mathbf{r}^a(z) + \rho^a(\cos\phi\,\mathbf{d}_1(z) + \sin\phi\,\mathbf{d}_2(z)), \tag{28}$$

with $\mathbf{r}^a$, $\mathbf{d}_1$, and $\mathbf{d}_2$ defined as in the main text. In *Cicconofri et al., 2020*, this surface is modelled as a continuous subject to active shear deformations. Here, we base our model on the explicit description of the individual MTs. Following *Hilfinger and Jülicher, 2008*, we suppose that the axonemal constraints confine MTs on the Ax surface at a fixed angular distance $\Delta\phi = 2\pi/9$ between each other. More formally, for $j = 1, \ldots, 9$, we define the centerline $\mathbf{r}^j$ of the $j$-th MT as $\mathbf{r}^j(s) = \mathbf{C}(s, \phi_j)$, where

$$\mathbf{C}(s,\phi) = \boldsymbol{\chi}(Z(s,\phi),\phi), \quad \text{and} \quad \phi_j = 2\pi(2-j)/9. \tag{29}$$

The function $Z(s,\phi)$ in *Equation 29* is defined (implicitly) via the equality

$$Z(S(z,\phi),\phi) = z \quad \text{where} \quad S(z,\phi) = \int_0^z \|\partial_z\boldsymbol{\chi}(z',\phi)\|\,dz'. \tag{30}$$

From the definitions above follows $\|\partial_s\mathbf{r}^j\| = 1$, so that MTs are indeed inextensible and $s$ is their arc length. Moreover, the Taylor expansion of $\mathbf{C}$ at the first order in $\rho^a$ gives the approximated formula (*Equation 5*), with $\gamma_1$ and $\gamma_2$ given by *Equation 6*.

We associate to the $j$-th MT an orthonormal frame along $\mathbf{r}^j$ given by the unit vectors

$$\mathbf{e}_3^j(s) = \partial_s\mathbf{r}^j(s), \quad \mathbf{e}_1^j(s) = \mathbf{N}(s,\phi_j), \quad \text{and} \quad \mathbf{e}_2^j(s) = \mathbf{e}_3^j(s) \times \mathbf{e}_1^j(s), \tag{31}$$

where $\mathbf{N} = \cos\phi\,\mathbf{d}_1(Z) + \sin\phi\,\mathbf{d}_2(Z)$ is the (outer) unit normal to the cylindrical surface. The unit vectors (*Equation 31*) determine MTs' cross-section orientations. The unit vectors $\mathbf{e}_1^j(s)$ and $\mathbf{e}_2^j(s)$ lie on the cross-section centered at $\mathbf{r}^j(s)$, while $\mathbf{e}_3^j(s) = \partial_s\mathbf{r}^j(s)$ is orthogonal to it. The (passive) elastic energy of the Ax is given by the sum of the MTs' elastic energies

$$\mathcal{W}_{pas}^a = \sum_{j=1}^9 \frac{1}{2}\int_0^L B^m\big((U_1^j(s))^2 + (U_2^j(s))^2\big) + C^m U_3^j(s)^2\,ds, \tag{32}$$

$$\text{where} \quad U_1^j = \partial_s\mathbf{e}_2^j \cdot \mathbf{e}_3^j, \quad U_2^j = \partial_s\mathbf{e}_3^j \cdot \mathbf{e}_1^j, \quad \text{and} \quad U_3^j = \partial_s\mathbf{e}_1^j \cdot \mathbf{e}_2^j$$

are the strains associated to the $j$-th MT, while $B^m$ and $C^m$ are the MTs' bending and twisting moduli (respectively). At the leading order approximation in $\rho^a$ we have

$$\mathbf{e}_1^j \approx \cos\phi_j\mathbf{d}_1 + \sin\phi_j\mathbf{d}_2, \quad \mathbf{e}_2^j \approx -\sin\phi_j\mathbf{d}_1 + \cos\phi_j\mathbf{d}_2, \quad \text{and} \quad \mathbf{e}_3^j \approx \mathbf{d}_3. \tag{33}$$

From *Equation 33* and *Equation 3* follows that $\mathcal{W}_{pas}^a$, as defined in *Equation 32*, can indeed be approximated by the right hand side of *Equation 2*, with $B^a = 9B^m$ and $C^a = 9C^m$.

We consider now MT sliding. Fixed a point $\mathbf{r}^{j+1}(s)$ on the $(j+1)$-th MT's centerline, we look for the nearest point to $\mathbf{r}^{j+1}(s)$ on the centerline $\mathbf{r}^j$ of the $j$-th MT. Such a point $\mathbf{r}^j(s^*)$ (we can think of it as a projection) lies at some arc length $s^*$, which depends on $s$. We write $s^* = \Pi_j(s)$. We then define the sliding $\sigma_j(s)$ as the difference of the two arc lengths $s$ and $\Pi_j(s)$. More formally,

$$\sigma_j(s) = s - \Pi_j(s), \quad \text{where} \quad \Pi_j(s) = \underset{\xi}{argmin}\|\mathbf{r}^{j+1}(s) - \mathbf{r}^j(\xi)\|. \tag{34}$$

The figure below illustrates the geometric idea behind *Equation 34*.

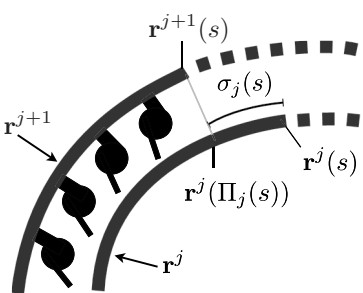

**Appendix 1—figure 1.** Sketch of two MTs' centerlines during deformation. The sliding $\sigma_j(s)$ is defined as the difference between the arc lengths $s$ and $\Pi_j(s)$. The latter is the arc length corresponding to the projection of $\mathbf{r}^{j+1}(s)$ on the curve $\mathbf{r}^j$. We have positive sliding when dyneins push the $j$-th MT toward the distal end of the flagellum and the $(j+1)$-th MT toward the proximal end.

The active internal energy of the Ax is defined as minus the total mechanical work of the dyneins

$$\mathcal{W}_{act}^a = -\sum_{j=1}^{9} \int_0^L F_j(s)\sigma_j(s)\,ds - \sum_{j=1}^{9} \widehat{F}_j \sigma_j(L)\,, \tag{35}$$

where $F_j(s)$ are the sliding forces on the $j$-th MT exerted by the dyneins on the $(j+1)$-th MT, and $\widehat{F}_j$ are the singular sliding forces (on the $j$-th MT exerted by the dyneins on the $(j+1)$-th MT) concentrated at the distal end of the Ax. Taylor expanding (*Equation 34*) in $\rho^a$ we have, at the leading order,

$$\sigma_j(s) \approx \rho^a(\cos\phi_{j+1} - \cos\phi_j)\gamma_1(s) + \rho^a(\sin\phi_{j+1} - \sin\phi_j)\gamma_2(s)\,, \tag{36}$$

with $\gamma_1$ and $\gamma_2$ given by *Equation 6*. From *Equation 36* we have that $\mathcal{W}_{act}^a$, as defined in *Equation 35*, is approximated by the right hand side of *Equation 4*, with

$$H_1(s) = \rho^a \sum_{j=1}^{9} (\cos\phi_{j+1} - \cos\phi_j)F_j(s)\,, \quad H_2(s) = \rho^a \sum_{j=1}^{9} (\sin\phi_{j+1} - \sin\phi_j)F_j(s)\,,$$

$$\widehat{H}_1 = \rho^a \sum_{j=1}^{9} (\cos\phi_{j+1} - \cos\phi_j)\widehat{F}_j\,, \quad \text{and} \quad \widehat{H}_2 = \rho^a \sum_{j=1}^{9} (\sin\phi_{j+1} - \sin\phi_j)\widehat{F}_j\,. \tag{37}$$

In the remainder we give some details of the derivation of *Equation 11* from *Equation 7*. Expanding *Equation 10* at the leading order in $\rho^p$ gives

$$\mathbf{g}_1(s) \approx \cos\phi^p \mathbf{d}_1(s) + \sin\phi^p \mathbf{d}_2(s)\,,$$

$$\mathbf{g}_2(s) \approx \frac{1}{\sqrt{1 + \gamma_{p\perp}(s)^2}}(-\sin\phi^p \mathbf{d}_1(s) + \cos\phi^p \mathbf{d}_2(s) + \gamma_{p\perp}(s)\mathbf{d}_3(s))\,,$$

$$\text{and} \quad \mathbf{g}_3(s) \approx \frac{1}{\sqrt{1 + \gamma_{p\perp}(s)^2}}(\gamma_{p\perp}(s)\sin\phi^p \mathbf{d}_1 - \gamma_{p\perp}(s)\cos\phi^p \mathbf{d}_2(s) + \mathbf{d}_3(s))\,, \tag{38}$$

with $\gamma_{p\perp}(s) = -\sin\phi^p \gamma_1(s) + \cos\phi^p \gamma_2(s)$. From *Equation 8* and *Equation 38*, leading order calculations give $V_1 \sim \rho^p$ and $V_3 \sim \rho^p$, whereas $V_2 \approx \gamma_{p\perp}/\sqrt{1 + \gamma_{p\perp}^2}$. Linearizing in $\gamma_{p\perp}$ we have $V_2 = \gamma_{p\perp}$, from which follows *Equation 11*.

## Appendix 2

### Dynamical equations

To derive *Equations 17–20,* it is convenient to introduce the quantities $M_1$, $M_2$ and $M_3$ defined via the following variational equality

$$\delta \mathcal{W} = \int_0^L M_1 \delta U_1 + M_2 \delta U_2 + M_3 \delta U_3 . \tag{39}$$

These quantities can be interpreted as the local components of the flagellar moment

$$\mathbf{M} = M_1 \mathbf{d}_1 + M_2 \mathbf{d}_2 + M_3 \mathbf{d}_3 . \tag{40}$$

A direct calculation gives

$$M_1(s) = B^a U_1(s) + \widehat{H}_2 + \int_s^L H_2 - D^p \cos \phi^p \int_s^L \gamma_{p\perp} ,$$
$$M_2(s) = B^a U_2(s) - \widehat{H}_1 - \int_s^L H_1 - D^p \sin \phi^p \int_s^L \gamma_{p\perp} , \quad \text{and} \quad M_3(s) = C^a U_3(s) \tag{41}$$

where $\gamma_{p\perp}(s) = -\sin \phi^p \gamma_1(s) + \cos \phi^p \gamma_2(s)$, as in the previous Section. We then write the variations $\delta U_i$ in terms of $\delta \theta_i$ (defined in the main text) obtaining

$$\delta U_1 = \partial_s \delta \theta_1 + \delta \theta_3 U_2 - \delta \theta_2 U_3 , \quad \delta U_2 = \partial_s \delta \theta_2 + \delta \theta_1 U_3 - \delta \theta_3 U_1 , \quad \delta U_3 = \partial_s \delta \theta_3 + \delta \theta_2 U_1 - \delta \theta_1 U_2 . \tag{42}$$

Combining *Equations 39, 40* and *Equation 42* we have

$$\delta \mathcal{W} = \mathbf{M}(L) \cdot \delta \boldsymbol{\theta}(L) - \int_0^L \partial_s \mathbf{M} \cdot \delta \boldsymbol{\theta} , \quad \text{and}$$
$$\delta \int_0^L \boldsymbol{\Lambda} \cdot (\partial_s \mathbf{r}^a - \mathbf{d}_3) = \boldsymbol{\Lambda}(L) \cdot \delta \mathbf{r}^a(L) - \int_0^L \partial_s \boldsymbol{\Lambda} \cdot \delta \mathbf{r}^a - \int_0^L (\mathbf{d}_3 \times \boldsymbol{\Lambda}) \cdot \delta \boldsymbol{\theta} .$$

In the calculations above, we took variations with $\delta \mathbf{r}^a(0) = \delta \boldsymbol{\theta}(0) = \mathbf{0}$, since we consider a flagellum with a clamped end at $s = 0$. Then, the principle of virtual work (*Equation 16*) yields the following force and torque balance equations

$$\partial_s \boldsymbol{\Lambda} + \mathbf{F} = \mathbf{0} \quad \text{and} \quad \partial_s \mathbf{M} + \mathbf{d}_3 \times \boldsymbol{\Lambda} + \mathbf{G} = \mathbf{0} , \tag{43}$$

$$\text{with} \quad \boldsymbol{\Lambda}(L) = \mathbf{0} , \quad \text{and} \quad \mathbf{M}(L) = \mathbf{0} . \tag{44}$$

*Equations 17–20* are derived from *Equation 43* and *Equation 44*, after some extra formal manipulations that we explain in the reminder.

We first introduce the local angular velocities

$$W_1 = \partial_t \mathbf{d}_2 \cdot \mathbf{d}_3 , \quad W_2 = \partial_t \mathbf{d}_3 \cdot \mathbf{d}_1 , \quad \text{and} \quad W_3 = \partial_t \mathbf{d}_1 \cdot \mathbf{d}_2 , \tag{45}$$

which are related to the strains via the following compatibility equations

$$\partial_t U_1 + U_2 W_3 - U_3 W_2 = \partial_s W_1 , \quad \partial_t U_2 + U_3 W_1 - U_1 W_3 = \partial_s W_2 , \quad \partial_t U_3 + U_1 W_2 - U_2 W_1 = \partial_s W_3 . \tag{46}$$

*Equation 46* follows from the identities $\partial_s \partial_t \mathbf{d}_i = \partial_t \partial_s \mathbf{d}_i$, with $i = 1, 2, 3$. We then rewrite the external forces and torques (*Equation 15*) in compact form as

$$\mathbf{F} = -\mathbf{V}[\mathbf{d}_3] \partial_t \mathbf{r}^a \quad \text{and} \quad \mathbf{G} = -\mu_r W_3 \mathbf{d}_3 , \quad \text{with} \quad \mathbf{V}[\mathbf{d}_3] = \mu_\perp (\mathbf{Id} - \mathbf{d}_3 \otimes \mathbf{d}_3) + \mu_{||} \mathbf{d}_3 \otimes \mathbf{d}_3 . \tag{47}$$

Scalar multiplying the torque balance equation by $\mathbf{d}_1$ and $\mathbf{d}_2$ we obtain the expressions for the first two local components

$$\Lambda_1 = -\partial_s \mathbf{M} \cdot \mathbf{d}_2 \quad \text{and} \quad \Lambda_2 = \partial_s \mathbf{M} \cdot \mathbf{d}_1 \tag{48}$$

of the Lagrange multiplier vector $\mathbf{\Lambda} = \Lambda_1 \mathbf{d}_1 + \Lambda_2 \mathbf{d}_2 + \Lambda_3 \mathbf{d}_3$. Scalar multiplying the torque balance equation by $\mathbf{d}_3$ gives

$$\mu_r W_3 = \partial_s \mathbf{M} \cdot \mathbf{d}_3 \,. \tag{49}$$

We rewrite the force balance equation as $\partial_t \mathbf{r}^a = \mathbf{V}[\mathbf{d}_3]^{-1} \partial_s \mathbf{\Lambda}$ and, after differentiating both sides with respect to $s$ and then scalar multiplying by $\mathbf{d}_1$, $\mathbf{d}_2$, and $\mathbf{d}_3$, we obtain

$$W_1 = -\partial_s \left( \mathbf{V}[\mathbf{d}_3]^{-1} \partial_s \mathbf{\Lambda} \right) \cdot \mathbf{d}_2 \,, \quad W_2 = \partial_s \left( \mathbf{V}[\mathbf{d}_3]^{-1} \partial_s \mathbf{\Lambda} \right) \cdot \mathbf{d}_1 \,, \tag{50}$$

$$\text{and} \quad 0 = \partial_s \left( \mathbf{V}[\mathbf{d}_3]^{-1} \partial_s \mathbf{\Lambda} \right) \cdot \mathbf{d}_3 \,. \tag{51}$$

From *Equation 41, 46*, and *Equation 49* we obtain *Equation 18* by first differentiating with respect to $s$ both sides of *Equation 49*, and then by linearizing the resulting equation. Similarly, exploiting also *Equation 48* this time, we differentiate with respect to $s$ and then linearize *Equation 50* to obtain *Equation 17*. Boundary conditions in *Equation 19* follow from *Equation 44*, whereas *Equation 20* are derived from

$$\partial_s \mathbf{\Lambda}(0, t) = \mathbf{V}[\mathbf{d}_3] \partial_t \mathbf{r}^a (0, t) = \mathbf{0} \,, \tag{52}$$

which follows from the fixed end condition $\mathbf{r}^a(0, t) = \mathbf{0}$.

Given $U_1$, $U_2$, and $U_3$ the orthonormal frame $\mathbf{d}_1$, $\mathbf{d}_2$, and $\mathbf{d}_3$ can be recovered by integrating the system of equations

$$\partial_s \mathbf{d}_1 = U_3 \mathbf{d}_2 - U_2 \mathbf{d}_3 \,, \quad \partial_s \mathbf{d}_2 = U_1 \mathbf{d}_3 - U_3 \mathbf{d}_1 \,, \quad \partial_s \mathbf{d}_3 = U_2 \mathbf{d}_1 - U_1 \mathbf{d}_2 \,, \tag{53}$$

which is derived from *Equation 3*. We can then recover the centerline $\mathbf{r}^a$ by integrating $\partial_s \mathbf{r}^a = \mathbf{d}_3$. The PFR centerline $\mathbf{r}^p$ and the orthonormal frame $\mathbf{g}_1$, $\mathbf{g}_2$, and $\mathbf{g}_3$ follow from *Equation 9* and *Equation 10*.

## Appendix 3

### Numerical simulations I: equilibria

We define the nondimensional variables for arc length $x$, time $y$, strains $u_i$, shear forces $h_i$, and concentrated shear forces $\hat{h}_i$ as follows

$$x = s/L, \quad y = t/T, \quad u_i = U_i/L^{-1}, \quad h_i = H_i/B^a L^{-2}, \quad \text{and} \quad \hat{h}_i = \widehat{H}_i/B^a L^{-1}. \tag{54}$$

The equilibrium *Equation 12*, with *Equation 22*, is solved by seeking for a minimizer of the energy (*Equation 23*) in nondimensional form

$$w = \frac{1}{2} \int_0^1 \left\| \mathbf{u} - \begin{pmatrix} u_1^* \\ u_2^* \end{pmatrix} \right\|^2 + \nu \left( \int_0^x \mathbf{e}_p \cdot \mathbf{u} \right)^2 - u_2^{*2}, \quad \text{where} \quad \nu = \frac{D^p}{B^a L^{-2}}.$$

In the formula above $\mathbf{u} = (u_1, u_2)$ is the nondimensional bending vector and $u_i^* = U_i^*/L^{-1}$, with $i = 1, 2$, are the nondimensional target bending strains. We find the minimizer using the gradient descent method

$$\mathbf{u}^{n+1} = \mathbf{u}^n - \alpha \frac{\delta w}{\delta \mathbf{u}}[\mathbf{u}^n], \quad \text{where}$$

$$\frac{\delta w}{\delta \mathbf{u}}[\mathbf{u}^n] = \mathbf{u}^n - \begin{pmatrix} u_1^* \\ u_2^* \end{pmatrix} + \nu \mathbf{e}_p \otimes \mathbf{e}_p \int_x^1 \int_0^{x'} \mathbf{u}^n(x'') dx'' dx'$$

and $\alpha$ is a conveniently chosen step size. We initiate the algorithm with $\mathbf{u}^0 = (u_1^*, u_2^*)$, and then we iterate it until $\|\mathbf{u}^{n+1} - \mathbf{u}^n\|$ falls below a pre-set tolerance parameter.

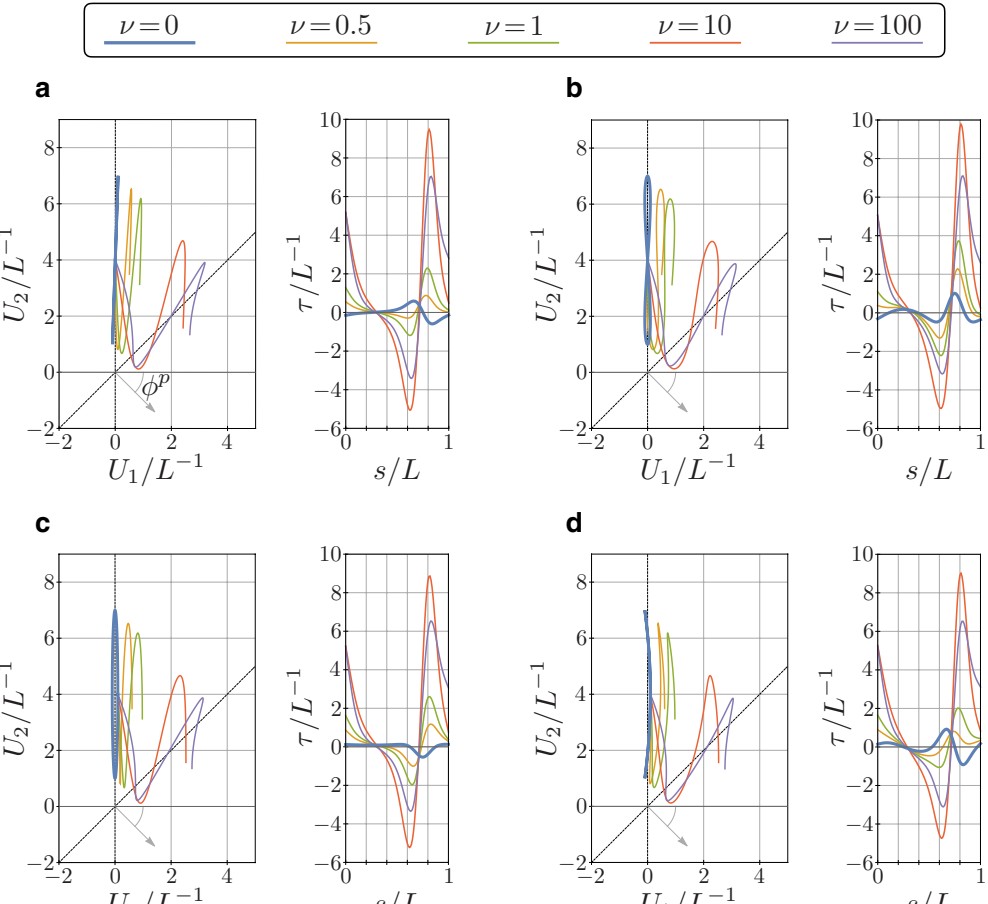

**Appendix 3—figure 1.** Bending vector traces and torsion of equilibrium configurations of the
*Appendix 3—figure 1 continued on next page*

euglenid flagellum under four different target bending strains, and for different values of the parameter $\nu = D^p/(B^a L^{-2})$. The target bending strains are given by $(u_1^*, u_2^*)$, with $u_2^*(x) = a_0 + a_1 \sin(2\pi x)$ and $u_1^* = \epsilon \tilde{u}_1^*$, where (a) $\tilde{u}_1^*(x) = \sin(2\pi x)$, (b) $\tilde{u}_1^*(x) = \sin(4\pi x)$, (c) $\tilde{u}_1^*(x) = \cos(2\pi x)$, and (d) $\tilde{u}_1^*(x) = \cos(4\pi x)$.

In the figure above, we considered four different perturbations of a perfectly planar, *Chlamydomonas*-like, target bending strain, and calculated the resulting flagellar equilibrium configurations for different values of the parameter $\nu$. Each configuration presents a different (non-null) torsional profile for $\nu = 0$ (thick, light blue lines). For larger values of $\nu$, all perturbations lead to (roughly) similar configurations, with torsional peaks of alternate sign.

## Appendix 4

### Numerical simulations II: flagellar dynamics

We consider the nondimensional variables defined in *Equation 54*. The dynamic *Equation 17* is recast, and then solved, in terms of the shear vector $\boldsymbol{\gamma} = (\gamma_1, \gamma_2)$ where

$$\gamma_1(x) = \int_0^x u_2\,, \quad \text{and} \quad \gamma_2(x) = -\int_0^x u_1\,. \tag{55}$$

After solving for $\boldsymbol{\gamma}$, we obtain the bending strains by differentiation. The equation for $\boldsymbol{\gamma}$ is

$$\eta\partial_y\boldsymbol{\gamma} = -\partial_x^4\boldsymbol{\gamma} + \partial_x^2\mathbf{h} + \nu\mathbf{e}_p^\perp \otimes \mathbf{e}_p^\perp\partial_x^2\boldsymbol{\gamma}\,, \quad \text{where} \quad \eta = \frac{\mu_\perp T^{-1}L^2}{B^a L^{-2}}\,, \tag{56}$$

and $\mathbf{h} = (h_1, h_2)$. The corresponding boundary conditions are given by

$$\partial_x\boldsymbol{\gamma}|_{x=1} + \hat{\mathbf{h}} = \mathbf{0}\,, \quad \left(\partial_x^2\boldsymbol{\gamma} - \mathbf{h} - \nu\mathbf{e}_p^\perp \otimes \mathbf{e}_p^\perp\boldsymbol{\gamma}\right)|_{x=1} = \mathbf{0}\,,$$

$$\boldsymbol{\gamma}|_{x=0} = \mathbf{0}\,, \quad \left(\partial_x^3\boldsymbol{\gamma} - \partial_x\mathbf{h} - \nu\mathbf{e}_p^\perp \otimes \mathbf{e}_p^\perp\partial_x\boldsymbol{\gamma}\right)|_{x=0} = \mathbf{0} \tag{57}$$

where $\hat{\mathbf{h}} = (\hat{h}_1, \hat{h}_2)$. The above (*Equation 57*) are point-wise conditions, which do not involve integral terms as in *Equation 19*. Avoiding this non-locality allows for an easier numerical implementation. The finite difference scheme we employ to solve *Equations 56–57* is illustrated in the remainder.

We consider the discrete time sequence $y_n = n\Delta y$ with $n = 0, 1, 2, \ldots$ and we define $\boldsymbol{\gamma}^n(x) = \boldsymbol{\gamma}(y_n, x)$, $\mathbf{h}^n(x) = \mathbf{h}(y_n, x)$, and $\hat{\mathbf{h}}^n = \hat{\mathbf{h}}(y_n)$. *Equation 56* is discretized in time with the one-step (semi-implicit) numerical scheme

$$\frac{\eta}{\Delta y}(\boldsymbol{\gamma}^{n+1} - \boldsymbol{\gamma}^n) = -\partial_x^4\boldsymbol{\gamma}^{n+1} + \partial_x^2\mathbf{h}^n + \nu\mathbf{e}_p^\perp \otimes \mathbf{e}_p^\perp\partial_x^2\boldsymbol{\gamma}^n\,, \tag{58}$$

and complemented by the boundary conditions

$$\partial_x\boldsymbol{\gamma}^{n+1}|_{x=1} + \hat{\mathbf{h}}^{n+1} = \mathbf{0}\,, \quad \left(\partial_x^2\boldsymbol{\gamma}^{n+1} - \mathbf{h}^n - \nu\mathbf{e}_p^\perp \otimes \mathbf{e}_p^\perp\boldsymbol{\gamma}^n\right)|_{x=1} = \mathbf{0}\,,$$

$$\boldsymbol{\gamma}^{n+1}|_{x=0} = \mathbf{0}\,, \quad \left(\partial_x^3\boldsymbol{\gamma}^{n+1} - \partial_x\mathbf{h}^n - \nu\mathbf{e}_p^\perp \otimes \mathbf{e}_p^\perp\partial_x\boldsymbol{\gamma}^n\right)|_{x=0} = \mathbf{0}\,. \tag{59}$$

The nondimensional arc length interval $[0, 1]$ is discretized uniformly in $M + 1$ points $x_k = k\Delta x$, with $k = 0, 1, \ldots, M = 1/\Delta x$. We also consider the extra 'ghost points' $x_k = k\Delta x$ with $k = -1, M+1, M+2$. The discrete values $\partial_x^2\boldsymbol{\gamma}_k^n = \partial_x^2\boldsymbol{\gamma}^n(x_k)$ of the second derivative in *Equation 56* are approximated by the finite difference scheme

$$\partial_x^2\boldsymbol{\gamma}_k^n = \begin{cases} (2\boldsymbol{\gamma}_0^n - 5\boldsymbol{\gamma}_1^n + 4\boldsymbol{\gamma}_2^n - \boldsymbol{\gamma}_3^n)/\Delta x^2 & \text{for } k = 0, \\ (\boldsymbol{\gamma}_{k-1}^n - 2\boldsymbol{\gamma}_k^n + \boldsymbol{\gamma}_{k+1}^n)/\Delta x^2 & \text{for } 1 \le k \le M - 1, \\ (-\boldsymbol{\gamma}_{M-3}^n + 4\boldsymbol{\gamma}_{M-2}^n - 5\boldsymbol{\gamma}_{M-1}^n + 2\boldsymbol{\gamma}_M^n)/\Delta x^2 & \text{for } k = M, \end{cases} \tag{60}$$

where $\boldsymbol{\gamma}_k^n = \boldsymbol{\gamma}^n(x_k)$. Analogous formulas are employed for the second derivative of $\mathbf{h}^n$. The discrete values $\partial_x^4\boldsymbol{\gamma}_k^{n+1} = \partial_x^4\boldsymbol{\gamma}^{n+1}(x_k)$ of the forth derivative in *Equation 56* are given by the scheme

$$\partial_x^4\boldsymbol{\gamma}_k^{n+1} = \begin{cases} (2\boldsymbol{\gamma}_{-1}^{n+1} - 9\boldsymbol{\gamma}_0^{n+1} + 16\boldsymbol{\gamma}_1^{n+1} - 14\boldsymbol{\gamma}_2^{n+1} + 6\boldsymbol{\gamma}_3^{n+1} - \boldsymbol{\gamma}_4^{n+1})/\Delta x^4 & \text{for } k = 0, \\ (\boldsymbol{\gamma}_{k-2}^{n+1} - 4\boldsymbol{\gamma}_{k-1}^{n+1} + 6\boldsymbol{\gamma}_k^{n+1} - 4\boldsymbol{\gamma}_{k+1}^{n+1} + \boldsymbol{\gamma}_{k+2}^{n+1})/\Delta x^4 & \text{for } 1 \le k \le M, \end{cases} \tag{61}$$

which involves the ghost points values $\boldsymbol{\gamma}_{-1}$, $\boldsymbol{\gamma}_{M+1}$, and $\boldsymbol{\gamma}_{M+2}$. The discretized approximations of the boundary conditions (*Equation 59*) are given by

$$\frac{\boldsymbol{\gamma}_{M+1}^{n+1} - \boldsymbol{\gamma}_{M-1}^{n+1}}{2\Delta x} = -\hat{\mathbf{h}}^{n+1}\,, \quad \frac{\boldsymbol{\gamma}_{M+2}^{n+1} - 2\boldsymbol{\gamma}_M^{n+1} + \boldsymbol{\gamma}_{M-2}^{n+1}}{4\Delta x^2} = \nu\mathbf{e}_p^\perp \otimes \mathbf{e}_p^\perp\boldsymbol{\gamma}_M^n + \mathbf{h}_M^n\,, \quad \boldsymbol{\gamma}_0^{n+1} = \mathbf{0}\,,$$

$$\frac{-3\boldsymbol{\gamma}_{-1}^{n+1} + 10\boldsymbol{\gamma}_0^{n+1} - 12\boldsymbol{\gamma}_1^{n+1} + 6\boldsymbol{\gamma}_2^{n+1} - \boldsymbol{\gamma}_3^{n+1}}{2\Delta x^3} = \nu\mathbf{e}_p^\perp \otimes \mathbf{e}_p^\perp\frac{-3\boldsymbol{\gamma}_0^n + 4\boldsymbol{\gamma}_1^n - \boldsymbol{\gamma}_2^n}{2\Delta x} + \frac{-3\mathbf{h}_0^n + 4\mathbf{h}_1^n - \mathbf{h}_2^n}{2\Delta x}\,.$$

The previous formulas give us the expressions for the ghost points' values $\boldsymbol{\gamma}_{-1}^{n+1}$, $\boldsymbol{\gamma}_{M+1}^{n+1}$, and $\boldsymbol{\gamma}_{M+2}^{n+1}$ in terms of $\hat{\mathbf{h}}^{n+1}$, $\boldsymbol{\gamma}_k^{n+1}$, $\boldsymbol{\gamma}_k^n$, and $\mathbf{h}_k^n$ with $0 \leq k \leq M$. These expressions are then plugged in *Equation 61*. In turn, the iterative scheme (*Equation 58*) allows to calculate $\boldsymbol{\gamma}_k^{n+1}$ from $\boldsymbol{\gamma}_k^n$ with $0 \leq k \leq M$, while incorporating the boundary conditions (*Equation 59*) in the numerical solution. The scheme is iterated for several time periods until a periodic solution is reached.

We obtain the history of shear forces presented in *Figure 7* by solving the following inverse dynamical problem. We assign first a history of normalized bending strains $(0, u_2(x, y))$, periodic in time, that imitates the experimentally observed *Chlamydomonas* flagellar curvatures reported in *Qin et al., 2015*. We use the following model

$$u_2(x, y) = a_0 - a_1 \cos(\lambda(x) - \omega(y))$$
$$\text{with} \quad \lambda(x) = 2\pi\lambda_0(x + \lambda_1 \sin(\pi x)) \quad \text{and} \quad \omega(y) = 2\pi(y + \omega_1 \sin(\pi y)). \tag{62}$$

Then, we calculate the shear forces that generate said history of bending strains for an Ax beating in a viscous fluid (without extra-axonemal structures attached to it). *Equation 56* with $\nu = 0$ and $h_2 = \widehat{h}_2 = 0$ defines exactly the dynamics of this system. We can solve for $h_1$ and $\widehat{h}_1$ explicitly, obtaining $\widehat{h}_1(y) = -u_2(1, y)$ and $h_1(x, y) = \partial_x u_2(x, y) - \eta \int_x^1 \int_0^{x'} \int_0^{x''} \partial_y u_2(x''', y)\, dx'''dx''dx'$.

In the dynamic simulation shown in *Figure 7*, we used the following numeric values for the physical parameters of the system. The bending modulus of the Ax is $B^a = 840 pN \cdot \mu m^2$, taken from *Xu et al., 2016*. We set $L = 28\mu m$, $T = 25ms$, and $\mu_\perp = 3.1 fN \cdot s \cdot \mu m^{-2}$, which are all values estimated in *Rossi et al., 2017*. The angle between spontaneous bending plane and the Ax-PFR joining line $\phi^p = -2\pi/9$ is estimated from micrographs in *Melkonian et al., 1982* and *Bouck et al., 1990*. For the bending strains parameters in *Equation 62* we took $a_0 = 7.8$, $a_1 = 7.5$, $\lambda_0 = 1.85$, $\lambda_1 = 0.1$, and $\omega_1 = -0.1$.

Without direct measurements for $D^p$, we set $\nu = 20$ as the value that best replicates the shapes and tridimensionality of the Euglena beat. The nondimensional parameter $\nu = D^p/B^a L^{-2}$ can be defined as the ratio between the elastic energy of an Ax bent with constant curvature $1/L$ and the elastic energy of the PFR deformed with uniform shear $\gamma = 1$. The two deformations do not happen simultaneously during flagellar deformation, where the kinematics of Ax and PFR are coupled. For this reason, the numerical value of parameter $\nu$ does not give a reliable estimate of the relative contribution ox Ax and PFR to the total elastic energy during flagellar deformation. One can define an effective parameter $\nu_e$ to measure the interplay of PFR and Ax elasticity in the following way. We can consider the elastic energy of an Ax bent along the spontaneous bending plane with constant curvature $1/L$, thus $U_1 = 0$ and $U_2 = 1/L$. This deformation of the Ax induces a PFR shear deformation with $\gamma_1 = s/L$ and $\gamma_2 = 0$ (due to the kinematic constraints, see 'Mechanical model' Section). In this case, the (passive) elastic energy of the Ax is given by $\mathcal{W}_{pas}^a = \frac{1}{2}B^a L^{-1}$, while the elastic energy of the PFR is given by $\mathcal{W}^p = \frac{1}{6}LD^p \sin^2(\phi^p)$. We define the effective parameter $\nu_e$ as the ratio between these two energies, which gives $\nu_e = \frac{1}{3}\sin^2(\phi^p)\nu$. For $\phi^p = -2\pi/9$, the value for the angle we used for *E. gracilis*, we have $\nu_e = 0.13\nu$. Thus, for $\nu = 20$ we have $\nu_e = 2.75$. This value of $\nu_e$ is close to one, showing that our simulations lead to large non-planarity of the flagellar beat when the two competing elastic energies of Ax and PFR are of similar magnitude. This is in agreement with our hypothesis that non-planarity emerges form competing and frustrated elastic structures, via the principle of structural incompatibility.

## Appendix 5

### Numerical simulations III: swimming

We associate to the (elliptical) cell body a moving frame of orthonormal unit vectors $\mathbf{i}$, $\mathbf{j}$ and $\mathbf{k}$, aligned with the major axis of the cell, see the figure below. The moving frame is located at the geometric center of the body $\mathbf{c}$. We denote with $\bar{\mathbf{r}}(s,t)$ the flagellum centerline in the moving frame coordinates. In the lab frame coordinates, the centerline of the flagellum is thus given by

$$\mathbf{r}(s,t) = \mathbf{c}(t) + \mathbf{Q}(t)\bar{\mathbf{r}}(s,t)\,,$$

where $\mathbf{Q}(t)$ is the cell body rotation, which has the column-wise matrix expression

$$\mathbf{Q}(t) = (\mathbf{i}(t)\,|\,\mathbf{j}(t)\,|\,\mathbf{k}(t))\,.$$

In our simulation, the (periodic) time history of curves $\bar{\mathbf{r}}(s,t)$ is given by the the flagellar shapes obtained from our model, shown in the 'Hydrodynamic simulations' Section of the main text. The body center trajectory $\mathbf{c}(t)$ and the rotation $\mathbf{Q}(t)$ are quantities to be obtained from the swimming simulation. We have

$$\partial_t\mathbf{r}(s,t) = \dot{\mathbf{c}}(t) + \boldsymbol{\omega}(t) \times \mathbf{Q}(t)\bar{\mathbf{r}}(s,t) + \mathbf{Q}(t)\partial_t\bar{\mathbf{r}}(s,t) = \mathbf{Q}(t)(\bar{\mathbf{v}}(t) + \bar{\boldsymbol{\omega}}(t) \times \bar{\mathbf{r}}(s,t) + \partial_t\bar{\mathbf{r}}(s,t))\,,$$

where $\boldsymbol{\omega}(t)$ is the rotational velocity of the cell body, $\bar{\mathbf{v}} = \mathbf{Q}^T\dot{\mathbf{c}}$, and $\bar{\boldsymbol{\omega}} = \mathbf{Q}^T\boldsymbol{\omega}$. From the force balance equations of the swimmer we can derive a formula for the quantities $\bar{\mathbf{v}}$ and $\bar{\boldsymbol{\omega}}$, which are completely determined by $\bar{\mathbf{r}}$ and the geometry of the cell body (which are given quantities). The hydrodynamic forces acting on the flagellum and the cell body are approximated using Resistive Force Theory (*Equation 15*). The linear density force $\mathbf{f}(s,t)$ at time $t$ exerted by the surrounding fluid can be written as

$$\mathbf{f}(s,t) = -\mu_\perp\mathbf{v}_\perp(s,t) - \mu_{||}\mathbf{v}_{||}(s,t) = -\mu_\perp\partial_t\mathbf{r}(s,t) - (\mu_{||} - \mu_\perp)\mathbf{v}_{||}(s,t)\,,$$

where $\mathbf{v}_{||} = \partial_s\mathbf{r}\,(\partial_s\mathbf{r} \cdot \partial_t\mathbf{r})$ and $\mathbf{v}_\perp = \partial_t\mathbf{r} - \mathbf{v}_{||}$ are the local velocity components parallel and perpendicular to the flagellum. We have

$$\mathbf{v}_{||}(s,t) = \mathbf{Q}\partial_s\bar{\mathbf{r}}\,(\partial_s\bar{\mathbf{r}} \cdot \bar{\mathbf{v}} + (\bar{\mathbf{r}} \times \partial_s\bar{\mathbf{r}}) \cdot \bar{\boldsymbol{\omega}} + \partial_s\bar{\mathbf{r}} \cdot \partial_t\bar{\mathbf{r}})\,.$$

The total viscous force $\mathbf{f}_{tot}$ and torque $\mathbf{g}_{tot}$ acting on the swimmer are given by

$$\mathbf{f}_{tot} = \mathbf{f}_{body} + \mathbf{f}_{flag} \quad \text{and} \quad \mathbf{g}_{tot} = \mathbf{g}_{body} + \mathbf{g}_{flag}\,,$$

where $\mathbf{f}_{flag}$ and $\mathbf{g}_{flag}$ are total viscous force and torque acting on the flagellum

$$\mathbf{f}_{flag} = \int_0^L \mathbf{f}(s,t)\,ds \quad \text{and} \quad \mathbf{g}_{flag} = \int_0^L (\mathbf{r}(s,t) - \mathbf{c}(t)) \times \mathbf{f}(s,t)\,ds\,,$$

while $\mathbf{f}_{body}$ and $\mathbf{g}_{body}$ are the total force and torque acting on the cell body

$$\mathbf{f}_{body} = \mathbf{Q}\mathbf{A}_{body}\bar{\mathbf{v}} \quad \text{and} \quad \mathbf{g}_{body} = \mathbf{Q}\mathbf{D}_{body}\bar{\boldsymbol{\omega}}\,, \quad \text{where}$$

$$\mathbf{A}_{body} = \begin{pmatrix} -A_\perp & 0 & 0 \\ 0 & -A_\perp & 0 \\ 0 & 0 & -A_{||} \end{pmatrix} \quad \text{and} \quad \mathbf{D}_{body} = \begin{pmatrix} -D_\perp & 0 & 0 \\ 0 & -D_\perp & 0 \\ 0 & 0 & -D_{||} \end{pmatrix}\,.$$

For the detailed expressions of the viscous resistive coefficients $A_{||}$, $A_\perp$, $D_{||}$, and $D_\perp$ see, for example *Kim and Karrila, 2013*. The force and torque balance on the cell gives us the equations of swimming dynamics

$$\mathbf{f}_{tot} = \mathbf{0} \quad \text{and} \quad \mathbf{g}_{tot} = \mathbf{0}\,,$$

which lead to the following formula

$$\begin{pmatrix} \bar{\mathbf{v}} \\ \bar{\boldsymbol{\omega}} \end{pmatrix} = -\begin{pmatrix} \mathbf{A} & \mathbf{B} \\ \mathbf{B}^T & \mathbf{D} \end{pmatrix}^{-1} \begin{pmatrix} \bar{\mathbf{f}} \\ \bar{\mathbf{g}} \end{pmatrix}\,,$$

where

$$\mathbf{A} = \mathbf{A}_{body} + \mathbf{A}_{flag}, \quad \mathbf{B} = \mathbf{B}_{flag}, \quad \mathbf{D} = \mathbf{D}_{body} + \mathbf{D}_{flag},$$

$$\mathbf{A}_{flag} = -\mu_\perp \mathbf{Id} - (\mu_\| - \mu_\perp) \int_0^L \partial_s \bar{\mathbf{r}} \otimes \partial_s \bar{\mathbf{r}}, \quad \mathbf{B}_{flag} = \mu_\perp \int_0^L [\bar{\mathbf{r}}]_\times - (\mu_\| - \mu_\perp) \int_0^L \partial_s \bar{\mathbf{r}} \otimes (\bar{\mathbf{r}} \times \partial_s \bar{\mathbf{r}}),$$

$$\mathbf{D}_{flag} = \mu_\perp \int_0^L [\bar{\mathbf{r}}]_\times^2 - (\mu_\| - \mu_\perp) \int_0^L (\bar{\mathbf{r}} \times \partial_s \bar{\mathbf{r}}) \otimes (\bar{\mathbf{r}} \times \partial_s \bar{\mathbf{r}}),$$

$$\bar{\mathbf{f}} = -\mu_\perp \int_0^L \partial_t \bar{\mathbf{r}} - (\mu_\| - \mu_\perp) \int_0^L \partial_s \bar{\mathbf{r}}(\partial_s \bar{\mathbf{r}} \cdot \partial_t \bar{\mathbf{r}}),$$

$$\bar{\mathbf{g}} = -\mu_\perp \int_0^L \bar{\mathbf{r}} \times \partial_t \bar{\mathbf{r}} - (\mu_\| - \mu_\perp) \int_0^L (\bar{\mathbf{r}} \times \partial_s \bar{\mathbf{r}})(\partial_s \bar{\mathbf{r}} \cdot \partial_t \bar{\mathbf{r}}).$$

In the identities above $[\mathbf{a}]_\times$ denotes the skew-symmetric matrix generated by the vector $\mathbf{a}$, which is defined as the matrix such that $[\mathbf{a}]_\times \mathbf{b} = \mathbf{a} \times \mathbf{b}$ for every vector $\mathbf{b}$. The time evolution of $\mathbf{c}(t)$ and $\mathbf{Q}(t)$ are then obtained by integrating the equations

$$\frac{d\mathbf{c}}{dt} = \mathbf{Q}(t)\bar{\mathbf{v}}(t) \quad \text{and} \quad \frac{d\mathbf{Q}}{dt} = \mathbf{Q}(t)[\bar{\boldsymbol{\omega}}(t)]_\times.$$

Results of the simulation are shown in *Figure 8* and in *Video 3*. In the figure below we show the calculated components of the vectors $\bar{\mathbf{v}} = (\mathbf{v} \cdot \mathbf{i}, \mathbf{v} \cdot \mathbf{j}, \mathbf{v} \cdot \mathbf{k})$ and $\bar{\boldsymbol{\omega}} = (\boldsymbol{\omega} \cdot \mathbf{i}, \boldsymbol{\omega} \cdot \mathbf{j}, \boldsymbol{\omega} \cdot \mathbf{k})$ during a flagellar beat. We considered an ellipsoidal cell $42\,\mu m$ long and $7.2\,\mu m$ wide, with a flagellum of length $L = 28\,\mu m$. The flagellum shapes are those illustrated in *Figure 7*, and we use for the resistive coefficients the values estimated in *Rossi et al., 2017*.

The simulations show a good qualitative agreement with the experimentally observed trajectories. They predict a (generalized) helical trajectory with radius $\approx 1.4\,\mu m$ and with the cell closing a full turn of the helix in $\sim 24$ beats, in excellent agreement with observations of *E. gracilis* specimens (*Rossi et al., 2017*). The pitch of the numerical trajectory is $\approx 9.5\,\mu m$, $\sim 0.2$ times the length of the cell which is less then typically observed values of $\sim 0.7$.

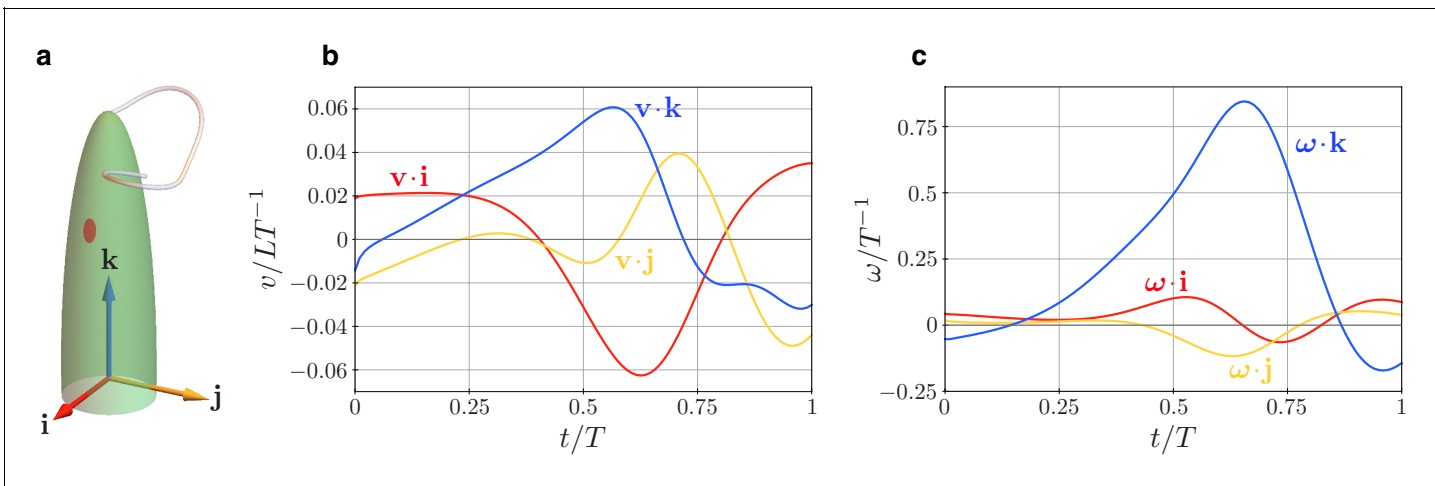

**Appendix 5—figure 1.** Swimming kinematics: numerical results. (a) Cell body and body frame unit vectors. (b) Body velocity and (c) rotational velocity projections on the body frame vectors during a flagellar beat.

The online version of this article includes the following source code for figure app51:

**Appendix 1—Source code 1.** Swimming dynamics solver.

