## [Decision Letter]

**Acceptance summary:**

The large photosynthetic biflagellate Euglena displays an intriguing swimming dynamics. The single emergent flagellum is held in front of the cell, where it executes a highly-3D, so-called spinning lasso beat pattern. Unusually, the euglena flagellum is flanked by a paraflagellar or para-axial rod structure, whose function is so far unknown. The authors put forward an interesting proposition (based on geometric mechanics) that a structural incompatibility between the two filaments leads to frustration and a torsional signature, consistent with the lasso. Both the static case and the hydrodynamic case were tested – the torsional feature seems to be robust to external forcing.

**Decision letter after peer review:**

Thank you for submitting your article "The biomechanical role of extra-axonemal structures in shaping the flagellar beat of Euglena" for consideration by *eLife*. Your article has been reviewed by three peer reviewers, one of whom is a member of our Board of Reviewing Editors, and the evaluation has been overseen by Aleksandra Walczak as the Senior Editor. The reviewers have opted to remain anonymous.

The reviewers have discussed the reviews with one another and the Reviewing Editor has drafted this decision to help you prepare a revised submission.

Summary:

The large photosynthetic biflagellate Euglena displays an intriguing swimming dynamics. The single emergent flagellum is held in front of the cell, where it executes a highly-3D, so-called spinning lasso beat pattern. Unusually, the euglena flagellum is flanked by a paraflagellar or para-axial rod structure, whose function is so far unknown. The authors put forward an interesting proposition (based on geometric mechanics) that a structural incompatibility between the two filaments leads to frustration and a torsional signature, consistent with the lasso. Both the static case and the hydrodynamic case were tested – the torsional feature seems to be robust to external forcing.

The reviewers found this to be a clearly written and well-explained paper that should be of interest to a wide range of researchers in various fields of biology and biophysics. While the level of mathematics is non-trivial, it is explained well within the text, and illustrated well in the figures.

Essential revisions:

1) The numbering of MTs is chosen by the authors does not appear to be consistent with published literature. This is important and worth checking carefully, in order to cross-reference any previous structural or EM studies. Usually, authors adopt a certain convention when it comes to numbering of microtubule doublets, as such systems do not exhibit perfect rotational symmetry but retain many heterogeneities – and there could be distinguished associations between certain Mts and other, non-axonemal structures such as mastigonemes.

2) According to the literature, e.g. Hyams, 1982, the PFR is hollow (Figure 3 of that paper). In contrast in *T. brucei* there are some longitudinal filaments and more like a lattice structure. Does this matter? The ventral flagellum also has a PFR – does this serve any purpose? Also, in euglena the rod extends all the way along the flagellum, but this is not the case in some other species. Do the authors believe that their conclusion extends across phyla, and if not, why not?

a) There is earlier work on the shape of spirochetes where a similar issue of two elastically competing structures appears: Dombrowski, et al., (2009).

The authors may wish to comment on the similarities and differences between that study and theirs.

3) In many other types of cilia and flagella which exhibit strongly non-planar beats, there is no association with a PFR-like structure, i.e. the latter is not necessary to generate non-planar bending. What's different here? Would be good to make and discuss some more cross-species comparisons (e.g. see Koyfman et al., 2011)

4) Is it clear that the rod assumes a consistent relationship with the numbered doublets? Could the rod itself be twisted around the axoneme? Authors assumed a fixed relationship and phi_p angle, and that this is neither perpendicular nor parallel to the beat plane… this is important according to Equations 25/26 (make sure to cite Melkonian, 1982 when mentioning the phi_p angle.)

5) In order to test their hypothesis the authors implemented a *Chlamydomonas* like planar beat pattern for dynein actuation, and determined the mismatch between the rod and axoneme… however, why could the naked axonemal beat not be 3D in the first place? Is there any structural evidence (as exists in *Chlamydomonas*) which suggests that the euglena beat is planar?

6) The authors have also assumed that the PFR, together with the AX_PFR linkages are passive. We suspect this is not the case. The fact that the PFR can be digested easily by trypsin suggests it may be contractile. The question is then could this structure participate in phototaxis – what is known about how the beat pattern changes during phototaxis – could the PFR actively enable a faster change in beat pattern or even beat frequency – via active contraction say, upon perception of photic signals? This could potentially be testable in the model.

7) Finally, the para-axial rod could also provide or promote a number of other functions, for instance gliding motility. What do the authors think about this possibility? Or conversely, can they suggest experiments that would help show more definitively that the PFR facilities 3D swimming?

8) The authors state (subsection “Mechanical model') that "Classical estimations on homogeneous elastic rods, see, e.g., Goriely, (2017), show that bending and twist moduli scale with the forth power of the cross section radius, whereas shear and stretching moduli scale with the second power and hence they are dominant for small radii." That there is a crossover is clear, but numbers matter here: For realistic material properties what is the length scale of the crossover?

(Also, "forth"->"fourth").

9) Although the mechanical description is general, the main claim of the paper that a sequence of shapes with the observed characteristics can emerge from a combination of a passive and active component with the given constrains is supported.

It would be interesting to know if the application of a hydrodynamic model to the theoretical waveforms can produce swimming paths similar to the experimentally observed ones.

10) The conclusion of the paper is of general nature, thus it could be noted that even in cilia with quasi-planar beats (like is the case for *Chlamydomonas*) asymmetric sliding constrains exist and that it is still debated if those are involved in generating axonemal twist or torsional waves during the beat. Therefore, it could be interesting to also discuss the effect of a weakly constrained beat 'weak para-flagella rod'.

How does the spinning-lasso beat breaks down?

What signature would waveforms under those conditions have?

---

## [Author Response]

Essential revisions:1) The numbering of MTs is chosen by the authors does not appear to be consistent with published literature. This is important and worth checking carefully, in order to cross-reference any previous structural or EM studies. Usually, authors adopt a certain convention when it comes to numbering of microtubule doublets, as such systems do not exhibit perfect rotational symmetry but retain many heterogeneities – and there could be distinguished associations between certain Mts and other, non-axonemal structures such as mastigonemes.

We have changed the numbering of MTs to make it consistent with published literature on *E. gracilis* EM studies. Specifically, we adopted the numbering convention of Melkonian et al., (1982) and Bouck et al., (1990). MT 1 is defined as the doublet with the most prominent attachment to the PFR. In this convention, MTs’ numbering increases in the clockwise direction when seen from the distal end of the Ax. This is at odds with the MTs’ labelling typically used in structural studies of cilia (which is the one we adopted in the first submission of our paper). In the typical case, MTs’ numbering increases in the anti-clockwise direction when seen from the distal end of the Ax. Dynein arms extend from MT N and reach MT N+1. In the convention adopted in this revised version of our manuscript, dynein arms extend from MT N and reach MT N-1, as in Figure 14 in Melkonian et al., (1982) and Figure 2 in Bouck et al., (1990).

2) According to the literature, e.g. Hyams, 1982, the PFR is hollow (Figure 3 of that paper). In contrast in *T. brucei* there are some longitudinal filaments and more like a lattice structure. Does this matter? The ventral flagellum also has a PFR – does this serve any purpose? Also, in euglena the rod extends all the way along the flagellum, but this is not the case in some other species. Do the authors believe that their conclusion extends across phyla, and if not, why not?

Our model for the PFR as a homogeneous elastic rod does not capture the level of detail that the reviewer is aiming for. We consider it as an “effective' coarse-grained model giving a fair first approximation of the forces of elastic origin that the PFR is subject to and transmits to the Ax. Through a suitable choice of geometric and material parameters, it can be used to model both hollow structures and structures with a dense core. Therefore, we believe that our conclusions are robust even against uncertainty of *some* structural details. See answer to no. 3 for more comments on this issue (in particular, comparisons across species).

The question about the ventral flagellum is interesting but beyond the scope of our work.

a) There is earlier work on the shape of spirochetes where a similar issue of two elastically competing structures appears: Dombrowski, et al., (2009).The authors may wish to comment on the similarities and differences between that study and theirs.

We thank the reviewers for pointing out the study by Dombrowski et al., to us. We have added it to our reference list, and commented on the similarities between the mechanism of shape-morphing in *B. burgdorferi* and that of flagellar beating in *E. gracilis*. Both mechanisms rely on mechanical “competition' between substructures, which generates frustration resulting in nontrivial shapes.

3) In many other types of cilia and flagella which exhibit strongly non-planar beats, there is no association with a PFR-like structure, i.e. the latter is not necessary to generate non-planar bending. What's different here? Would be good to make and discuss some more cross-species comparisons (e.g. see Koyfman et al., 2011)

Yes, the association with a PFR-like structure is not necessary to generate non-planarity in flagella. We did not imply otherwise, but we have made this point clearer in our resubmission. Our main conclusion applies to *E. gracilis* specifically, where a prominent asymmetry in the PFR-Ax bonding complex is present. In our model, the PFR-Ax interaction is needed to generate the typical spinning lasso geometrical signature which defines the peculiar Euglena beating style.

In other PFR-bearing flagella, like those of *Leishmania mexicana* and *Trypanosoma brucei*, such a prominent asymmetry is lacking, see e.g. Portman and Gull, (2010). For these organisms, the angle ϕp is closer to zero, as we mentioned in our first submission. In this case, our model predicts a planar (or quasi planar) beat and this prediction is consistent with the observed behaviour of both *L. mexicana* and *T. brucei* mutants with body-detached flagellum. Following the suggestion by the reviewer, we have added a reference to Koyfman et al., (2011).

4) Is it clear that the rod assumes a consistent relationship with the numbered doublets? Could the rod itself be twisted around the axoneme? Authors assumed a fixed relationship and phi_p angle, and that this is neither perpendicular nor parallel to the beat plane… this is important according to Equations 25/26 (make sure to cite Melkonian, 1982 when mentioning the phi_p angle.)

We believe that the EM study by Melkonian et al., (1982) gives a clear answer to the question raised by the reviewer. The possibility of a systematic numbering convention of MTs strongly implies a consistent relationship between Ax and PFR. The authors define a scheme for the flagellar cross-section “based on the evaluation of 80 different cross-sections of flagella', where MTs are identified based on associations with PFR and mastigonemes. The bonding links complex is conserved through the length of the flagellum (and across flagella), and it seems hard to reconcile this observation with the possibility that the PFR was twisted around the Ax. We cited Melkonian et al., (1982) in the paper when mentioning the ϕp angle.

5) In order to test their hypothesis the authors implemented a Chlamydomonas like planar beat pattern for dynein actuation, and determined the mismatch between the rod and axoneme… however, why could the naked axonemal beat not be 3D in the first place? Is there any structural evidence (as exists in Chlamydomonas) which suggests that the euglena beat is planar?

In our model the role of the PFR is twofold: it acts as (1) a MTs’ sliding inhibitor, and (2) as a mechanically antagonistic element.

The Ax-PFR bonding links play the central (structural) role in MTs’ sliding inhibition. Our working hypothesis is that MTs’ sliding stretches the bonding links. This inhibits MTs’ sliding and triggers a dynein organization (via mechanical feedback) similar to the one present in *Chlamydomonas* and sperm cells. However, we are not yet able to model the microscopic details of this response mechanism. We take this feedback-based self-organization process as a given, and we consider a force pattern that would produce planar beat in absence of antagonistic forces on the Ax.

The agreement of the observed flagellar shapes with the ones we predict shows that our working hypothesis is consistent with the available evidence. Further work will be needed in the future to address the issue of whether the dynein actuation pattern is truly planar and, if so, which feedback mechanisms lead to the emergence of this pattern. We have clarified this point in the revised manuscript (see subsection “Dyneins’ actuation induced by sliding inhibition').

6) The authors have also assumed that the PFR, together with the AX_PFR linkages are passive. We suspect this is not the case. The fact that the PFR can be digested easily by trypsin suggests it may be contractile. The question is then could this structure participate in phototaxis – what is known about how the beat pattern changes during phototaxis – could the PFR actively enable a faster change in beat pattern or even beat frequency – via active contraction say, upon perception of photic signals? This could potentially be testable in the model.

This is an interesting and widely open question, but we believe that the answer is highly non-trivial. The question deserves further ad-hoc study, including a large set of new experiments and a modelling effort to address the mechanisms of active contraction, light perception, and their interplay. We consider this suggestion as a very interesting direction to pursue in future studies, and we have indicated so in Discussion section.

7) Finally, the para-axial rod could also provide or promote a number of other functions, for instance gliding motility. What do the authors think about this possibility? Or conversely, can they suggest experiments that would help show more definitively that the PFR facilities 3D swimming?

This is another interesting question, but we are not able to suggest an answer.

In other systems, the question of whether the paraflagellar rod is “vital' for a certain form of motility has been address with the study of mutants (see e.g., Bastin et al., 1998).Genetic editing techniques are not available to us, and we are unaware of results of this nature in the general literature on *E. gracilis* (possibly because its genome has not been sequenced yet).

Our experience with physical experiments on *E. gracilis* concerns swimming motility and metaboly, and we have no experience with gliding motility. We have not been able (at least not yet) to device a single physical experiment proving or disproving that PFR facilitates 3D swimming.

8) The authors state (subsection “Mechanical model') that "Classical estimations on homogeneous elastic rods, see, e.g., Goriely, (2017), show that bending and twist moduli scale with the forth power of the cross section radius, whereas shear and stretching moduli scale with the second power and hence they are dominant for small radii." That there is a crossover is clear, but numbers matter here: For realistic material properties what is the length scale of the crossover?

Consider the bending energy Wb and the shear energy Ws of a deformed homogeneous rod with circular cross-sections of radius ρ. Suppose that external forces induce a bending of the rod with curvature 1/L, and a shear γ. The bending and shear energies are given by Wb=12BL−1 and Ws=12LDγ2, respectively. The bending stiffness B and the shear stiffness D are given by

B=Yπρ44 and D=Y πρ22(1+p)χ,

where Y is the Young modulus of the rod, p is the Poisson ratio, and χ=3227 is a geometric coefficient. The bending energy is negligible with respect to the shear energy if WsWb≫1. For the PFR of *E. gracilis* we have L∼25μm and ρ∼80nm, see, e.g., Walne and Dawson, (1993). Thus, for every value of the Young modulus, and for a Poisson ratio within the typical range of 0 and ½, we have WsWb∼105γ2. Even for exceedingly small shear angles γ∼0.03 the strain energy dominates over the bending energy by two orders of magnitude WsWb∼102. For larger (and more realistic) shear angles this dominance increases.

(Also, "forth"->"fourth").

We have corrected the typo.

9) Although the mechanical description is general, the main claim of the paper that a sequence of shapes with the observed characteristics can emerge from a combination of a passive and active component with the given constrains is supported.It would be interesting to know if the application of a hydrodynamic model to the theoretical waveforms can produce swimming paths similar to the experimentally observed ones.

We have followed this suggestion. We have shown that the sequence of flagellar shapes obtained from our model (shown in subsection “Hydrodynamic simulations and comparison with observations', Figure 7) can generate swimming paths similar to those observed in experiments. Swimming cell simulation results are reported in Figure 8 in the main text, Appendix 5, and in Video 3. The results are briefly reviewed in the Discussion section, and more details on the implementation and physics behind swimming simulations are given in Appendix 5.

10) The conclusion of the paper is of general nature, thus it could be noted that even in cilia with quasi-planar beats (like is the case for Chlamydomonas) asymmetric sliding constrains exist and that it is still debated if those are involved in generating axonemal twist or torsional waves during the beat. Therefore, it could be interesting to also discuss the effect of a weakly constrained beat 'weak para-flagella rod'.How does the spinning-lasso beat breaks down?What signature would waveforms under those conditions have?

We have performed a sensitivity analysis of flagellar shapes with spontaneous *Chlamydomonas*-like bending under different perturbations, see Appendix 3. We relaxed the planar constraint and considered four different weakly non-planar spontaneous configurations. Each configuration presents a different (non-null) torsional profile when the antagonistic mechanical interaction of the PFR is absent ν=0. For larger values of ν, each perturbation assumes the typical spinning lasso geometry (torsional peaks of alternate sign), regardless of the native (ν=0) torsional profile. This shows that the PFR-Ax interaction strongly influence the shape outcome even when the perfectly planar constraint on the spontaneous bending is relaxed.